# Electric double layer-mediated polarization field for optimizing photogenerated carrier dynamics and thermodynamics

Chengxin Zhou[1], Jian Gao [1,2] ✉, Yunlong Deng[1], Ming Wang[1], Dan Li[1] & Chuan Xia[2] ✉

Photocatalytic hydrogen evolution efficiency is limited due to unfavorable carrier dynamics and thermodynamic performance. Here, we propose to introduce electronegative molecules to build an electric double layer (EDL) to generate a polarization field instead of the traditional built-in electric field to improve carrier dynamics, and optimize the thermodynamics by regulating the chemical coordination of surface atoms. Based on theoretical simulation, we designed CuNi@EDL and applied it as the cocatalyst of semiconductor photocatalysts, finally achieved a hydrogen evolution rate of 249.6 mmol h$^{-1}$ g$^{-1}$ and remained stable after storing under environmental conditions for more than 300 days. The high H$_2$ yield is mainly due to the perfect work function, Fermi level and Gibbs free energy of hydrogen adsorption, improved light absorption ability, enhanced electron transfer dynamics, decreased HER overpotential and effective carrier transfer channel arose by EDL. Here, our work opens up new perspectives for the design and optimization of photosystems.

Photocatalytic hydrogen evolution by water splitting is considered to be an important strategy to solve the energy and environmental crisis[1-10]. However, the solar-to-hydrogen (STH) efficiency is quite low due to the unsatisfactory carrier dynamics and thermodynamic performance, which is mainly caused by the easy recombination of photogenerated electrons-holes (e$^-$/h$^+$), high hydrogen evolution overpotential, and poor conductivity in semiconductor photosystems[11-17]. Previous studies have shown that the above problems can be alleviated by supporting cocatalysts and construct heterojunction to form build-in electric fields[18-27]. The noble metal Pt has been recognized as the most ideal hydrogen evolution reaction (HER) cocatalyst[28-38]. On the other hand, Li et al. designed S-scheme heterojunction WO$_3$/CoP and found that the photogenerated electron-hole recombination of CoP was prevent[2]. Jiang et al. fabricated ternary junction PTC/CdS/MIL-101 and found that PTC (polyoxo-titanium cluster) can promote the electron-hole separation of CdS[10]. Jing et al. designed a dual-porphyrin heterostructure of ZnTCPP/THPP (tetrakis (4-carboxyphenyl) zinc porphyrin/tetrakis (4-hydroxyphenyl) porphyrin) and obtained a high photocatalytic H$_2$ evolution rate of 41.4 mmol h$^{-1}$ g$^{-1}$ by loading with 6 wt% Pt as the cocatalyst, which is mainly attributed to the giant interfacial electric field formed between dual porphyrins greatly facilitates efficient charge separation and transfer[20]. However, the high cost and scarcity of Pt limit its application, moreover, the traditional built-in electric field has very limited promotion effect on carrier transport and the thermodynamic performance (Gibbs free energy of hydrogen adsorption ($\Delta G_{H^*}$)) is also unsatisfactory[39-50].

To improve the carrier dynamics and decrease the Gibbs free energy of hydrogen adsorption, here, we deliberately constructing a polarization field by structuring an electrical double layer (EDL) with the negative electrode outside between the surface and sub-surface, which is differ from the traditional build-in electric field. According to the theoretical analysis, introducing electronegative molecules on the

[1]New Energy Materials Laboratory, Sichuan Changhong Electronic (Group) Co.; Ltd., Chengdu 610041, China. [2]School of Materials and Energy, University of Electronic Science and Technology of China, Chengdu 611731, China. ✉e-mail: gaojian@changhong.com; chuan.xia@uestc.edu.cn

surface of CuNi to construct the EDL with a negative electrode outside was proposed. To this end, the effects of electronegative groups (such as carbonyl, ester, ect.) on CuNi in terms of work function, Fermi level, Gibbs free energy of hydrogen adsorption, and d-band center have been studied by Density Functional Theory (DFT) calculations. Results show that ester group (-O-C = O) can effectively increase the work function (Φ = 6.82 eV), optimize the chemical coordination of surface atoms, which is evidenced by the decrease the Fermi level and Gibbs free energy of hydrogen adsorption ($\Delta G_{H^*}$ = −0.039 eV). Based on the guidance of DFT, a solvothermal-calcination process was designed to synthesize the ester-modified CuNi (CuNi@O-C = O, CN@EDL). Subsequently, CN@EDL was used as the cocatalyst of the semiconductor photosystems. Photocatalytic tests show that the optimal CN@EDL/CdS affords $H_2$ evolution rate of 249.6 mmol h$^{-1}$g$^{-1}$, which is outperform than previous reports. Finally, the working mechanism of EDL on HER is deeply investigated by systematic experimental characterizations.

## Results

### Theoretical calculation results

DFT (Density Function Theory) calculations were performed firstly to demonstrate the conjecture that electronegative moleculars (such as -C = O and -O-C = O) can regulate the work function, H adsorption/desorption ability, d-band center, etc. The slab models are described in Supplementary Fig. 1. The distances between the carbon atoms that bond to the metal atoms are measured to be 2.474 Å and 2.737 Å for Cu-C and Ni-C respectively, which are larger than that of the Cu-C and Ni-C chemical bond lengths (1.8 - 2.0 Å). This proves that the functional group is attached to the surface of CuNi alloy in the form of free radicals, rather than forming Cu$_x$C compound.

### Work function, charge density difference, and density of states

As depicted in Supplementary Fig. 2, Fig. 1a and Supplementary Table 1, the work functions of CdS, CuNi, CuNi@C = O, and CuNi@O-C = O are calculated to be 5.16, 4.90, 6.04, and 6.82 eV, respectively. Obviously, the functional groups have increased the work function and decreased the Fermi level of CuNi, especially the ester group (-O-C = O). Electrons tend to transfer from a material with smaller work

function to one with larger work function. Consequently, the CuNi@O-C = O possesses a larger work function and lower Fermi level should capture electrons more easily[13].

The band structures and orbital compositions of the catalysts were calculated to investigate the electronic behavior (Supplementary Fig. 3a–f, Supplementary Fig. 4a and Fig. 1b). Depicted in Supplementary Fig. 3c–f, Supplementary Fig. 4a and Fig. 1b are the band structures and PDOS (projected density of states) of CuNi, CuNi@C = O, and CuNi@O-C = O respectively, which all have metallic characteristics and high conductivity due to the absence of a band gap. Specially, the more localized electron distribution and higher density of states of CuNi@O-C = O indicate higher electrical conductivity than CuNi and CuNi@C = O.

In order to analyze the formation of the EDL, charge density difference and Bader charge are calculated. Charge density difference diagrams were shown in Fig. 1c and Supplementary Fig. 4b–d, where cyan indicates decreased charge density and yellow indicates increased charge density. Bader charge calculations were performed to further analyzing the specific charge transfer amount (Fig. 1d). Markedly, the Cu and Ni atoms lose electrons, while O atoms get electrons, which is consistent with the calculation results of charge density difference.

According to the results of charge density difference and Bader charge analysis, it can be concluded that the EDL with a negative electrode outside is formed, and its working principle is depicted in Fig. 1e. The electrons tend to gather on the electronegative ester groups to form an additional polarization field pointing to the outside. It can be concluded that the introduction of the ester group capable of leading to the aggregation of electrons and form the EDL with negative electrode outside.

### D-band center calculation

The molecule adsorption ability pays credit to the intrinsic properties of d-orbitals of the transition metals, primarily on account of the interaction between the orbital of molecular and the d-orbital of metal will lead to the energy level splitting, as depicted in Supplementary Fig. 5, meanwhile the position of the formed anti-bond orbital is paramount for the stability of the material[7,51–57].

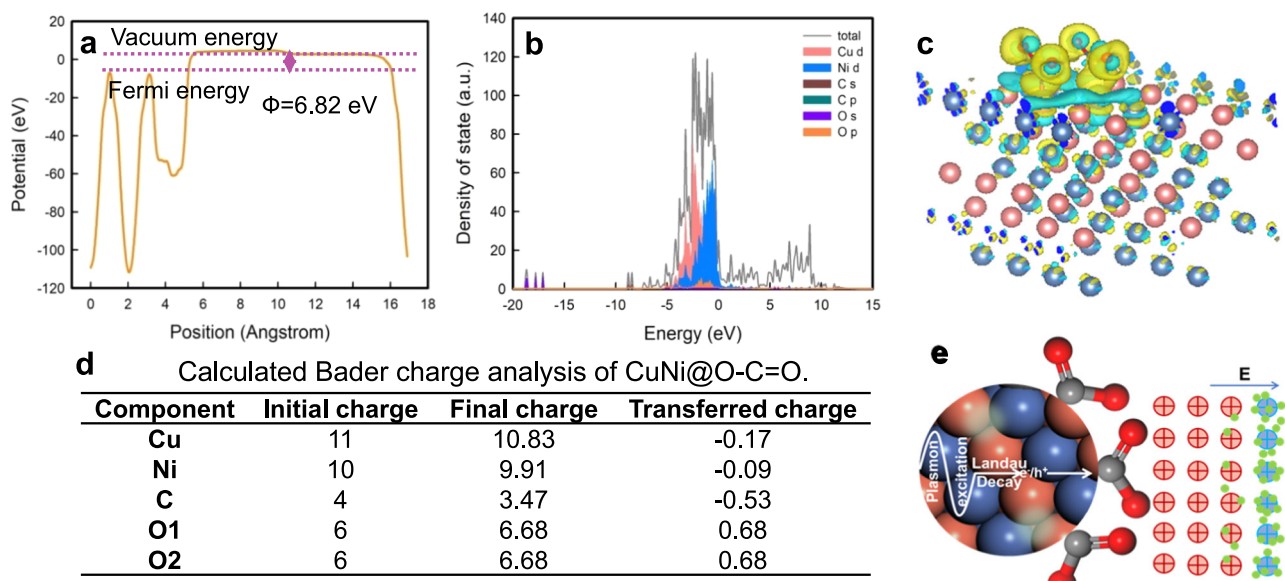

**d** Calculated Bader charge analysis of CuNi@O-C=O.

| Component | Initial charge | Final charge | Transferred charge |
|---|---|---|---|
| **Cu** | 11 | 10.83 | -0.17 |
| **Ni** | 10 | 9.91 | -0.09 |
| **C** | 4 | 3.47 | -0.53 |
| **O1** | 6 | 6.68 | 0.68 |
| **O2** | 6 | 6.68 | 0.68 |

**Fig. 1 | DFT calculation results of electronic characteristics and EDL mechanism. a** Work function and **b** the partial density of states of CuNi@O-C = O. **c** Charge density difference diagram of CuNi@O-C = O, where cyan indicates that the charge density decreases and yellow indicates that it increases. **d** The calculated Bader charge of CuNi@O-C = O. **e** Working mechanism of the EDL.

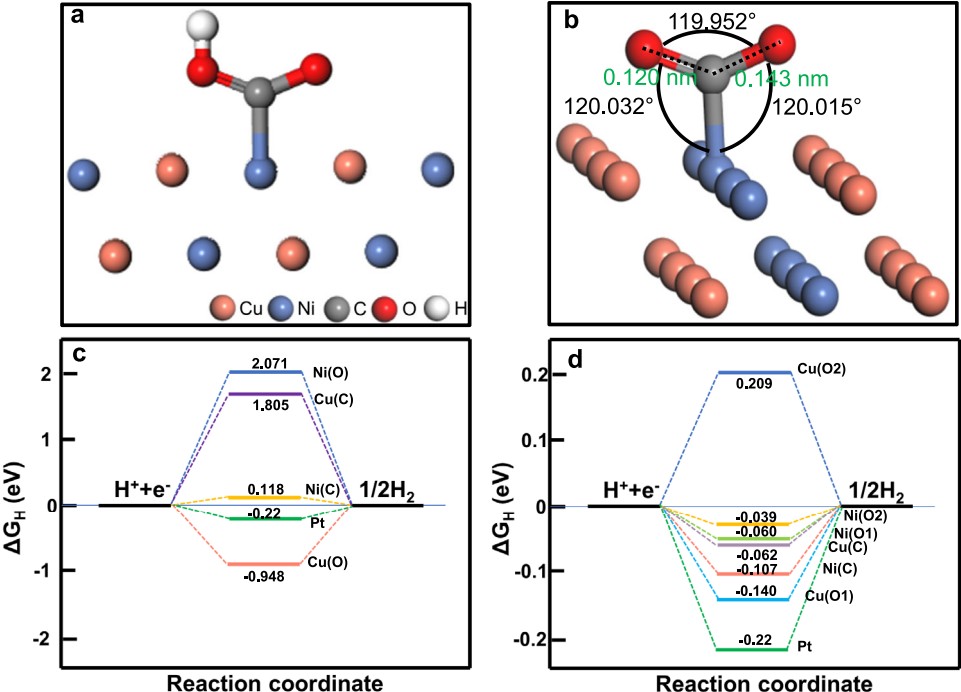

**Fig. 2 | H adsorption/desorption characteristics calculated by DFT. a** The slab models (side view) of the CuNi@O-C = O (111) plane with H adsorbed on O2 (Ni-O2). **b** Angle and bond length in ester group. **c, d** The calculated free energy diagrams for H$_2$ production: (**c**) CuNi@C = O and (**d**) CuNi@O-C = O.

The d-band center can be obtained according to PDOS results:

$$\varepsilon_d = \frac{\int_{-\infty}^{\infty} n_d(\varepsilon)\varepsilon d\varepsilon}{\int_{-\infty}^{\infty} n_d(\varepsilon) d\varepsilon} \qquad (1)$$

As shown in Supplementary Table 2, The d-band center of CuNi@O-C = O is more negative, indicating the elevation of H adsorption performance, which is conducive to HER.

### H adsorption energy and Gibbs free energy calculation

Simultaneously, the adsorption of H* and desorption of H$_2$ are essential for the hydrogen evolution rate, therefore, the H adsorption energies ($E_{ads}$) and Gibbs free energies ($\Delta G_H$) of CuNi, CuNi@C = O, CuNi@O-C = O were calculated, respectively (Supplementary Fig. 6–8, Fig. 2, Supplementary Table 3). The $\Delta G_H$ of Ni-O2 is closest to zero (−0.039 eV, even better than Pt, $\Delta G_{H(Pt)} = −0.22$ eV)[9], theoretically indicating that it is the optimal reaction site. On the whole, the $\Delta G_H$ of CuNi@O-C = O is relatively smaller, indicating that the chemical coordination of surface atoms was optimized and -O-C = O is more conducive to H adsorption and H$_2$ desorption.

### Experimental results

On the basis of the DFT results, the cocatalytic performance of CuNi will realize substantial improvement by introducing the ester group to form an EDL in terms of increasing the work function and H adsorption/desorption ability, reducing the Fermi level and d-band center[9,13,19,36,45]. Guided by theoretical calculations, we rationally designed a synthesis routine for CuNi@O-C = O (CN@EDL) and applied it as the cocatalyst of CdS. Ultimately, systematic experiments and characterizations were performed to uncover the structural properties and working mechanisms.

### Structure and composition analysis

The synthesis routine is shown in Supplementary Fig. 9. XRD tests were carried out first to study the composition and crystal form (Fig. 3a and Supplementary Figs. 10–12). As shown in Fig. 3a, the diffraction peaks

of CN@EDL are located between Cu (PDF#04-0836) and Ni (PDF#04-0850), indicating the formation of the CuNi alloy phase (PDF#65-9048) and its average particle size is calculated to be 17.4 nm according to Scherrer formula. For CN@EDL/CdS, the signals assigned to hexagonal CdS (PDF#41-1049) and CuNi can be retrieved, indicating the compounding of CN@EDL and CdS.

Next, SEM tests were investigated to study the structure and morphology of the catalysts. As shown in Fig. 3b and Supplementary Fig. 13a, b, CdS represents the snowflake structure formed by multiple pinnate leaves, in which the included angles between stem and stem, stem and leaf are both 60°. Meanwhile, hexagonal nanosheets attached to the stem have been found, further indicating the hexagonal phase properties of CdS (Supplementary Fig. 13c). Since CdS mainly exposes the (001) facet, the crystal will grow along [1–100] and the symmetry directions[1,10,27]. Consequently, the growth diagrams of CdS with pinnate and snowflake structures have been drawn (Supplementary Fig. 13d).

As illustrated in Supplementary Fig. 14 and Fig. 3c are the SEM spectra and partly enlarged view of CN@EDL/CdS, there are dense nanoparticles with a size of tens of nanometers attached to CdS, which are considered to be CN@EDL. The structure of CN@EDL is further investigated in Supplementary Fig. 15 and Fig. 3d, which is nanoparticle shaped with amorphous flocs wrapped on the surface, and the wrapping layer is preliminarily judged to be amorphous functional groups. In combination with XRD results, there is no diffraction peak of carbon, indicating that the groups are attached to the surface of CuNi alloy without carbon layer. As shown in Fig. 3e is the STEM image and element mapping of CN@EDL, which verifies the existence of Cu, Ni, C, and O elements. As shown in Supplementary Fig. 16a, b are the HRTEM image and SAED pattern of CdS. The lattice spacing of 0.36 nm corresponds to the (001) plane of CdS (PDF#41-1049) and the angle of lattice fringes (60°) is consistent with the results of XRD and SEM. The spots in the SAED pattern are clear and hexagonal, indicating that the hexagonal CdS was well crystallized. Supplementary Fig. 16c–f are the TEM and HRTEM images of CN@EDL/CdS. The lattice spacing of 0.20 nm corresponds to the (111) plane of CuNi (PDF#65-9048) and the

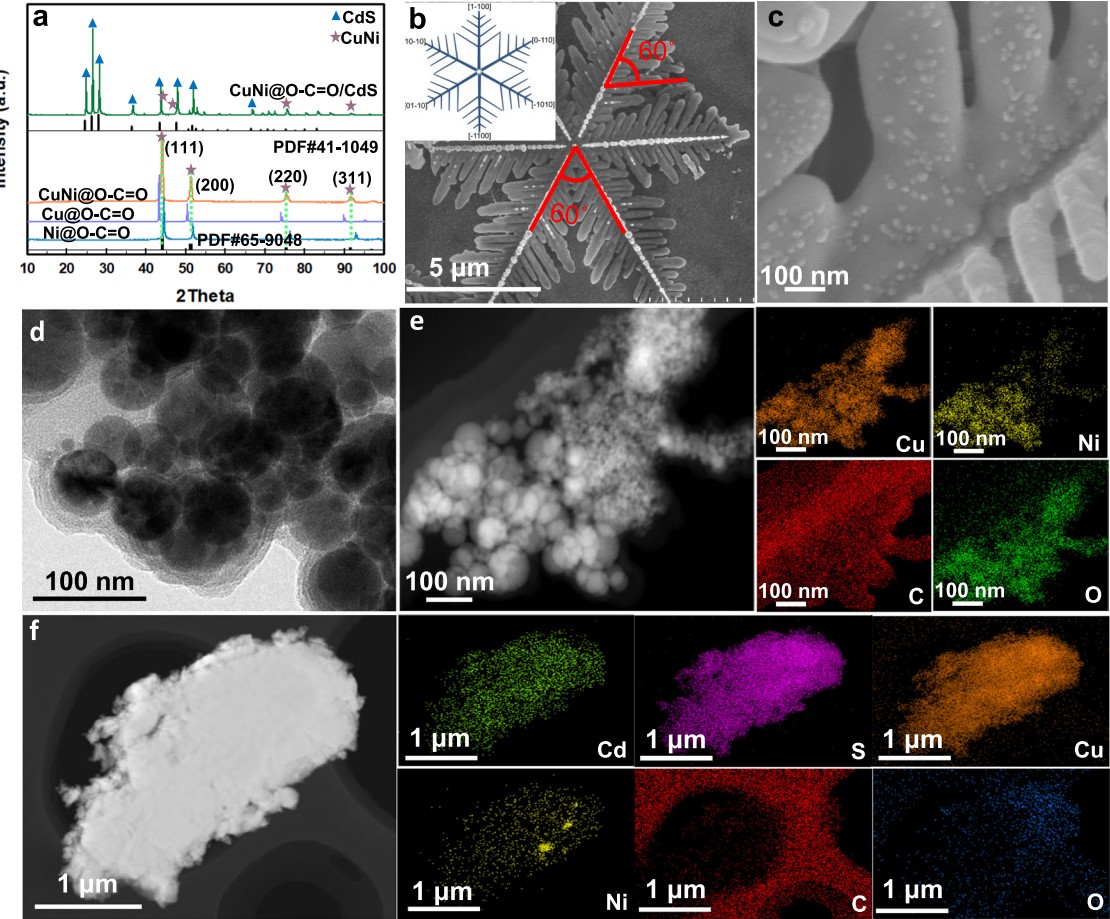

**Fig. 3 | Crystallinity and structure characterizations. a** XRD spectra of Cu@EDL, Ni@EDL, CN@EDL and CN@EDL/CdS. SEM patterns of **b** CdS (the insert is the growth crystal orientation of snowflake-like CdS), **c** CN@EDL/CdS. **d** TEM, **e** STEM-EDS element mapping images of CN@EDL. **f** STEM-EDS element mapping images of CN@EDL/CdS.

obvious boundary of the lattice regions (marked in blue) indicates the formation of heterojunction between CN@EDL and CdS. The STEM image and element mapping of CN@EDL/CdS are depicted in Fig. 3f, markedly, there are Cd, S, Cu, Ni, C, and O elements. From what has been discussed above, it can be concluded that CN@EDL/CdS has been successfully synthesized.

To further investigate the valence states of the elements, the type of molecular group, and the interaction between different components, XPS measurements were conducted (Supplementary Fig. 17–20 and Fig. 4). The peaks at 284.8, 286.2 and 288.6 eV of C 1 $s$ correspond to C-C, C = O and O-C = O configurations, respectively (Fig. 4a). While the peaks at 533.4, 531.9, 531.5 and 529.9 eV of O 1 $s$ can be assigned to adsorbed oxygen (O$_{ads}$), C-O, C = O and lattice oxygen (O$_{lat}$), respectively (Supplementary Fig. 18d). Therefore, it can be preliminarily verified that the ester groups (O-C = O) have been introduced to the surface of CuNi alloy.

FT-IR and Raman measurements were conducted to further verify the type of the surficial molecular groups (Fig. 4b, c and Supplementary Fig. 21). As shown in Fig. 4b, the FT-IR absorption peaks located at 916, 1109, 1363, 1726, and 3421 cm$^{-1}$ are corresponded to -CH, -C–O-C, -OH, -C = O and H$_2$O, respectively. The absorption peaks located at 400 - 900 cm$^{-1}$ (582 cm$^{-1}$, 638 cm$^{-1}$, 727 cm$^{-1}$) are correspond to Metal-O and Metal-O-C[18,20,25]. In particular, the peaks at 1726 cm$^{-1}$ and 1109 cm$^{-1}$ correspond to $\upsilon_{C=O}$ and $\upsilon_{C-O-C}$ of ester (-COOR), respectively, which are the characteristic signals of ester group (-O-C = O). As shown in Fig. 4c is the Raman spectrum, the signals located at 800, 1361, 1603, 2323, and 3147 cm$^{-1}$ are corresponded to -C–O-C, -C-C, -C = O, -CH, and

-OH, respectively, which indicates the existence of ester group. Meanwhile, the single at 127 cm$^{-1}$ corresponds to the lattice vibrations in crystals.

Subsequently, in order to study the interaction between CN@EDL and CdS, the XPS spectra of CN@EDL, CdS, and CN@EDL/CdS were compared. Specifically, the binding energies of Cd 3d in CN@EDL/CdS are positively shifted relative to that of pure CdS, and so are S 2$p$ (Supplementary Fig. 20). Conversely, the corresponding peaks of Cu 2$p$, Ni 2$p$, and C 1 $s$ are negatively shifted (Fig. 4d–f), proving that the junction is formed between them and that electrons migrate from CdS to CN@EDL. Concretely, CN@EDL is working as the electron acceptor and is the electron-rich site in the composite, which is consistent with the DFT results that CN@EDL with a higher work function can effectively capture electrons.

### Activity and stability of photocatalytic H$_2$ evolution

The photocatalytic HER experiments were conducted under visible light irradiation ($\lambda > 420$ nm). Shown in Fig. 5a, b, Supplementary Figure 22 and Supplementary Figure 23 are the hydrogen evolution rates and stabilities of CdS, CuNi/CdS, Pt/CdS(0.5 wt%, 1 wt%, 3 wt%, and 5 wt%), Cu@EDL/CdS, Ni@EDL/CdS, and CN@EDL/CdS(1:2, 1:1 and 2:1), respectively. After loading with CuNi, the HER rate of CdS increased from 7.0 mmol h$^{-1}$ g$^{-1}$ to 17.0 mmol h$^{-1}$ g$^{-1}$, while it increased to 108.5 mmol h$^{-1}$ g$^{-1}$ when loaded with noble metal Pt (1 wt%, which is highest in comparison with the loading contents of Pt are 0.5 wt%, 3 wt%, and 5 wt%), Markedly, it increased to 249.6 mmol h$^{-1}$ g$^{-1}$ when loaded with CN@EDL, which is even much superior that Pt/CdS,

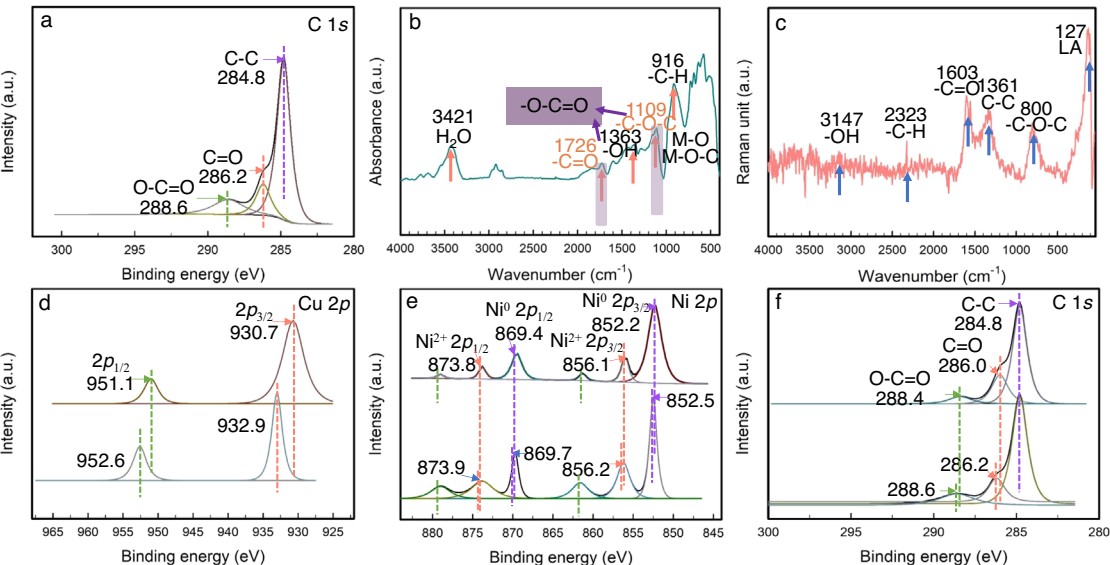

**Fig. 4 | Functional group and interfacial interaction analysis.** High-resolution XPS spectra of **a** C of CN@EDL. **b** FT-IR and **c** Raman spectra of CN@EDL. **d**–**f** High-resolution XPS spectra of (**d**) Cu, (**e**) Ni, and (**f**) C of samples before and after compounding.

indicating that CN@EDL is a more effective cocatalyst that Pt. The $H_2$ production yields and stabilities of different mass fractions of CN@EDL (2%, 5% and 8%) in CN@EDL/CdS were studied (Supplementary Figure 24), and the CN@EDL(5%)/CdS is the best-performing one, which reaches remarkable 35 times enhancement over the benchmark CdS, and much higher than that reported in the literature (Supplementary Table 4). The apparent quantum efficiency (AQE) of CN@EDL/CdS calculated by **Formula (S-5)** is 64.3% (Monochromatic light, $\lambda = 420$ nm). To avoid accidental experiments, five parallel samples were tested under the same condition. The photocatalytic activities were almost identical (Supplementary Fig. 25). The $H_2$ production rates were not attenuated after 16 h and five cycles (Fig. 5b and Supplementary Fig. 26). As shown in Supplementary Fig. 27, the XRD spectrum of CdS after reaction and placed for several days has changed markedly compared with that of fresh sample. And the peak appeared at 169.0 eV in the XPS spectrum of used CdS corresponding to S element with higher valence, indicating that part of $S^{2-}$ in CdS has been oxidized (Supplementary Fig. 28). Conversely, there was no noticeable difference in the XRD, XPS spectra and SEM patterns of fresh and used CN@EDL/CdS (Supplementary Fig. 29–33), indicating that the photocorrosion of CdS was effectively inhibited after loading CN@EDL. What's more, the photocatalytic hydrogen production, XRD and XPS spectra of CN@EDL/CdS stored for more than 300 days were also tested, results show that the photocatalysts were stable (Supplementary Fig. 34–36).

**Enhancement mechanism of hydrogen production activity**
Primarily, optical property and energy band structure are important factors that determine photo-quantum conversion efficiency. Therefore, we firstly investigated the UV-Vis absorption abilities of CdS, CN@EDL and CN@EDL(2%, 5%, 8%)/CdS, as shown in Fig. 5c and Supplementary Fig. 37. Markedly, the light absorption intensity of CN@EDL is noticeable higher than that of CdS. Furthermore, there are local surface plasmon resonance (LSPR) absorption peaks at 256 nm and 524 nm, as a result, there will be not only photoelectrons ($h^+$-$e^-$) produced by CdS, but also high-energy hot electrons (p-$e^-$) generated by CN@EDL (Fig. 5j). Specifically, the absorption intensity of CdS in the range of 530 to 800 nm is virtually zero, and the absorption edge ($\lambda_{max}$) at 508 nm corresponds to the absorption wavelength of the essential band gap ($E_g$) ($\lambda_{max} = 1240/E_g$). After loading CN@EDL, the absorption intensity of CdS system is enhanced in the range of 200

to 800 nm. From $(\alpha h\nu)^2$ to $h\nu$ plots in Fig. 5d and **Formula (S-1)**, the band gaps of CdS and CN@EDL/CdS are 2.44 and 2.43 eV, respectively.

To further investigate the conduction band (CB) and valence band (VB) potential, the Mott-Schottky (M-S) curves were obtained according to **Formula (S-2)** and **(S-3)**. As shown in Fig. 5e, f, the CB potentials of CdS and CN@EDL(5%)/CdS are −0.68 V and −0.88 V (vs RHE, pH = 0) ($E_{CB} = E_{FB}$−0.2), respectively (the M-S plots of CN@EDL(2%)/CdS and CN@EDL(8%)/CdS are shown in Supplementary Fig. 38). We therefore concluded that the reducibility of electrons in CN@EDL/CdS is stronger after loading CN@EDL. The VB potentials of CdS and CN@EDL/CdS are 1.76 and 1.55 eV respectively according to VB-XPS tests (Fig. 5g). The VB potential ($E_{VB}$) can be also obtained by **Formula (S-5)**. Therefore, the samples' specific values of $E_g$, $E_{FB}$, CB, and VB potentials were obtained, as shown in Supplementary Table 5.

The carrier density ($N_D$) is another critical factor determining the quantum efficiency. According to **Formula (S-3)**, the $N_D$ in CdS is calculated to be $4.6 \times 10^{20}$ cm$^{-3}$, while it is increased to $2.2 \times 10^{21}$ cm$^{-3}$ in CN@EDL/CdS. It has been shown that a collective oscillation of free carriers (LSPR effect) occurs when the carrier density reaches the order of $10^{21}$ cm$^{-3}$. That is, CN@EDL can effectively promote the separation of $e^-$/$h^+$ in CdS, which increases the electron density and leads to the emergence of LSPR effect (Fig. 5h).

The energy band structure and reaction mechanism diagrams were drawn (Fig. 5i). Firstly, when CdS is excited by light, the bounded electrons in VB will absorb photon energy ($h\nu$) and transfer to CB to become free electrons. As CN@EDL has a larger work function and a lower Fermi level than CdS, electrons tend to transfer from CdS to CN@EDL, thus reducing the $e^-$/$h^+$ recombination. Subsequently, the reduction of proton will occur on the surface of CN@EDL, which is conducive to the adsorption of H* and the desorption of $H_2$, as the DFT results show that the $\Delta G_H$ of CN@EDL is closer to zero.

Another important factor affecting the performance is the dynamics of photogenerated carriers, to this end, PL, TRPL, and photoelectrochemical tests (LSV, EIS) were performed. From the steady-state PL spectra (Fig. 5j), a main emission band is observed at ~550 nm, which is attributed to the band-to-band transition of CdS. Interestingly, an apparent fluorescence quenching was appeared when loaded with CN@EDL, especially for CN@EDL(5%)/CdS, proving that the carrier recombination in CdS system was inhibited.

Furthermore, the carriers' lifetime was investigated by time-resolved transient PL spectroscopy (TRPL), and the decay spectra were

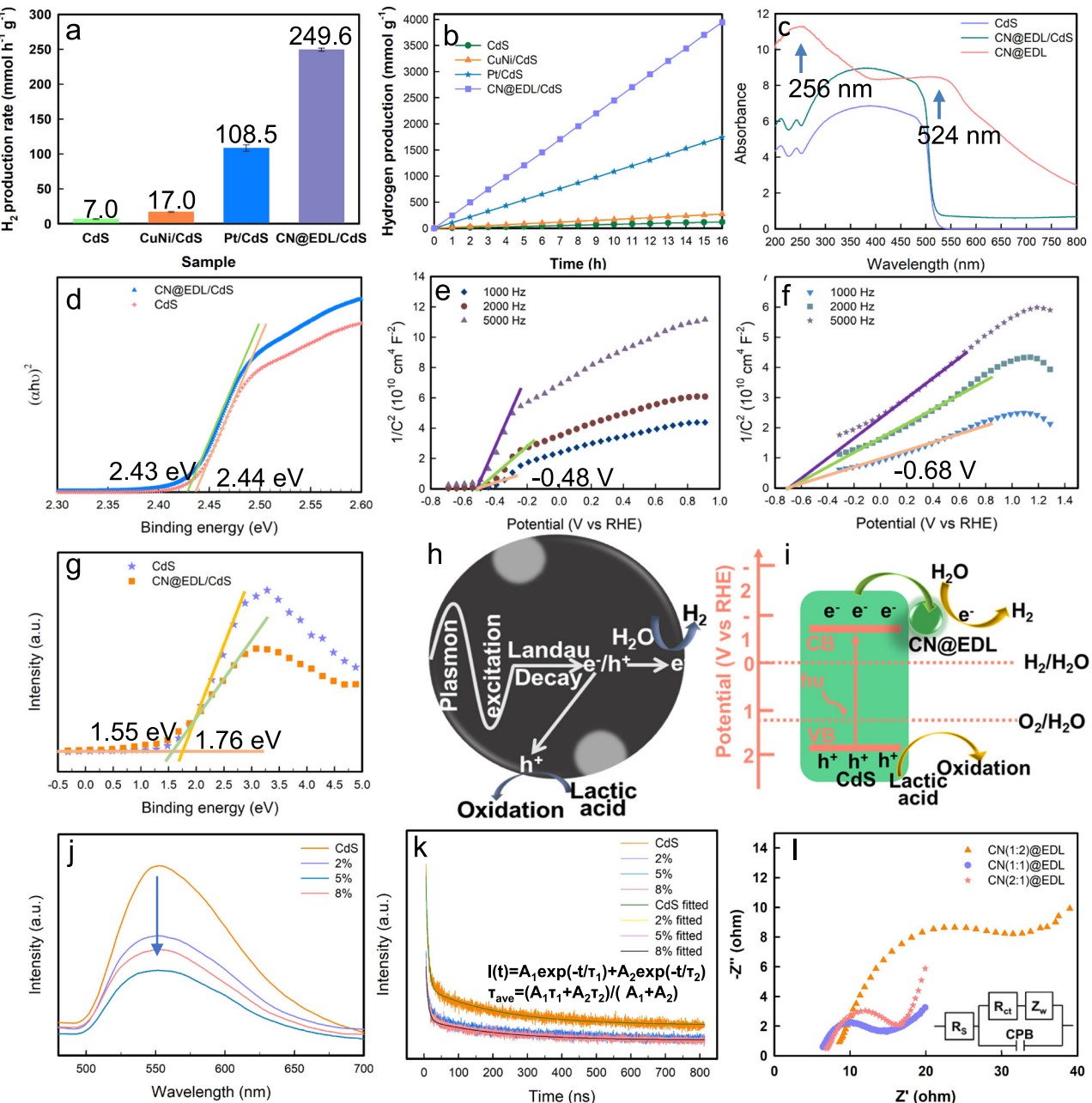

**Fig. 5 | Photocatalytic activity and mechanism research. a, b** Hydrogen production (**a**) rates and (**b**) stabilities of CdS, CuNi/CdS, Pt/CdS, and CN@EDL/CdS. Error bars in **a** are standard error values of three tests ($n = 3$). **c** UV-Vis absorption spectra of CdS, CN@EDL, and CN@EDL/CdS. **d** Tauc plots of $(\alpha h\nu)^2$ versus $h\nu$ of CdS and CN@EDL/CdS. **e, f** Mott-Schottky curves of CdS and CN@EDL/CdS tested at different frequencies (1000 Hz, 2000 Hz, and 5000 Hz), respectively. **g** VB-XPS spectra of CdS and CN@EDL/CdS. **h**, LSPR effect in CN@EDL and h-e⁻ generation progress. **i** Energy band diagram of CN@EDL/CdS. **j–l** Dynamic performance characterization: **j, k** Steady-state PL spectra (**j**) and TRPL decay curves (**k**) of CdS and CN@EDL/CdS (2%, 5%, 8%) under an excitation wavelength of 390 nm and observed at 550 nm. **l** EIS spectra of CN@EDL with different Cu and Ni ratios.

well fitted by the double-exponential decay model (Fig. 5k): $I(t)$ $=A_1\exp(-t/\tau_1)+A_2\exp(-t/\tau_2)$. The average carrier lifetime of the photocatalysts is calculated by the average emission attenuation intensity life formula ($\tau_{ave}=(A_1\tau_1+A_2\tau_2)/(A_1+A_2)$) and the fitting results are displayed in Supplementary Table 6. The average carrier lifetimes of (2%, 5%, 8%) CN@EDL/CdS are shorter than that of CdS, which indicates the formation of an effective carrier transfer channel between CN@EDL and CdS, thus improving the carrier transfer dynamics[11]. This conclusion was further confirmed by EIS experiments.

As shown in Fig. 5l, Supplementary Fig. 39 and Supplementary Table 7, Cu:Ni(1:1) possesses a much smaller EIS radius of 10.49 Ohm

demonstrating its excellent electrical conductivity. After loading CN@EDL, the EIS radius of CdS has been reduced from 121400 Ohm to 48730 Ohm, indicating improved conductivity and electron transition kinetics.

As shown in Supplementary Fig. 40, the HER overpotential of CN@EDL at current densities of 10 mA cm⁻² in a neutral electrolyte (0.5 M Na₂SO₄) is only 34 mV, while in acidic (0.5 M H₂SO₄) and alkaline (1 M KOH) electrolyte are 10 mV and 17 mV, respectively. It is comparable to that of noble metal Pt, which is paramount for improving the kinetics of HER. Conversely, the HER overpotential of CdS is −1.048 V, indicating an unsatisfactory hydrogen release kinetic of

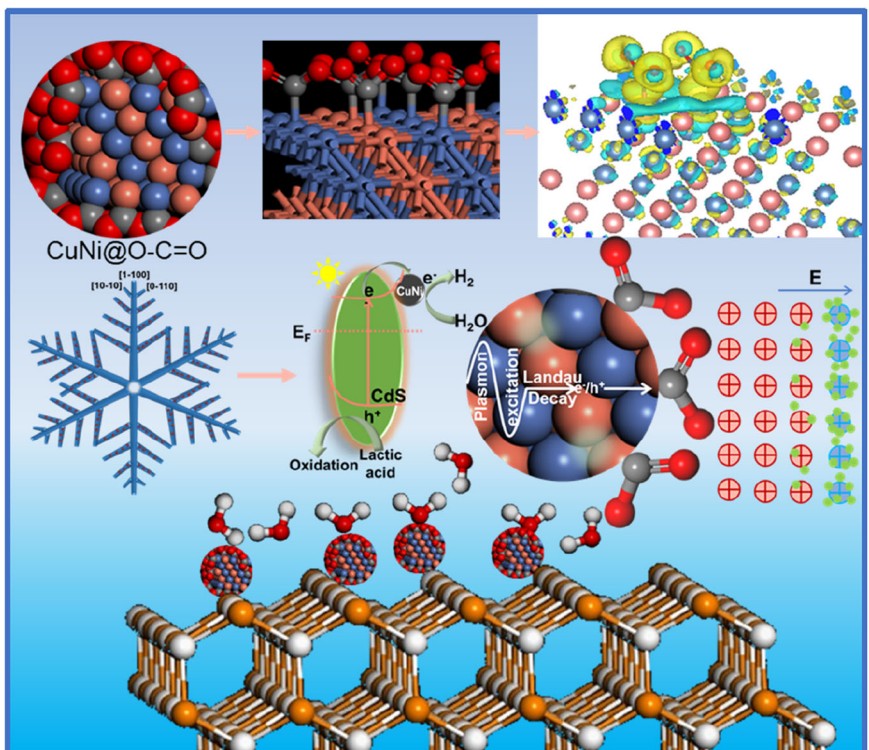

**Fig. 6 | Schematic diagram of microstructure and reaction mechanism.** The working mechanism of ester functional group, electric double layer, and polarization field. Combining with the DFT calculation results of charge density difference and Bader charge, the working mechanism of the EDL can be obtained. The electronegative ester group interacts with CuNi to form an electric double layer with the negative electrode outside, then, the CN@EDL nanoparticles are attached to the surface of CdS as a co-catalyst.

semiconductor material. After loading CN@EDL, it reduced to −0.674 V, suggesting the greatly accelerated kinetics (Supplementary Fig. 41).

Briefly, we have proposed a novel strategy to design efficient non-Pt cocatalysts from the molecular level by introducing electronegative groups to construct an EDL with the negative electrode outside. The polarization field would weaken the escape of electrons, thereby increasing the potential barrier of electrons transition from Fermi level to Vacuum level and eventually lead to the increasement of work function. DFT calculations were firstly conducted and the results indicate that ester group (-O-C = O) can effectively increase the work function and decrease the H adsorption free energy. Charge density difference and Bader charge calculations indicate that the ester group is an electronegative group with the ability to capture electrons from CuNi. Subsequently, the rationalized fabrication routine was applied to synthesize CuNi@EDL (CN@EDL) and the $H_2$ evolution rate of CN@EDL/CdS reached 249.6 mmol $h^{-1}$ $g^{-1}$. Finally, related characterizations were performed and found that the light absorption of CN@EDL is outstanding and the LSPR absorption in the visible light region can lead to the generation of hot electrons ($h-e^-$) except for the photogenerated electrons ($p-e^-$). The light absorption edge of CdS is blue-shifted and the CB potential is negatively shifted after loading CN@EDL, thereby improving the quantum efficiency and electron reduction ability.

To sum up, the main reasons for the excellent performance of CN@EDL/CdS can be summarized as follows: (1) The introduction of electronegative ester groups formed an EDL with a negative electrode outside on the surface of CuNi, which effectively increased the work function thus enhancing the ability to capture electrons and reduce the HER overpotential; (2) The ester groups optimized the H adsorption/desorption ability; (3) The outstanding light absorption and LSPR effect of CN@EDL led to the formation of additional high-energy hot electrons; (4) An effective carrier transfer channel was formed in the novel MIS photovoltaic devices CN@EDL/CdS, which ultimately improved the separation of $e^-/h^+$ (Fig. 6).

## Discussion

In summary, we have proposed a strategy to introduce electronegative molecules to build an electric double layer (EDL) to generate a polarization field instead of the traditional built-in electric field to improve carrier dynamics and optimize the chemical coordination of surface atoms, then, a series of ester-modified non-Pt HER cocatalysts with the deliberately structured polarization field were rationally designed and applied as the cocatalyst of snowflake CdS. The optimum CN@EDL(5%)/CdS exhibits the HER rate as high as 249.6 mmol $h^{-1}$ $g^{-1}$ and the apparent quantum yield 63% at 420 nm. The optimized work function, Fermi level, and Gibbs free energy of hydrogen adsorption, improved light absorption ability, enhanced electron transfer dynamics, decreased HER overpotential and effective carrier transfer channel arose by EDL are essential for interfacial carrier kinetics and surface reaction dynamics, as proved by experimental and computational investigations. This study proposes a new strategy for designing high-efficient non-Pt cocatalyst and photosystem with polarization field for various applications. Notably, the ester group may not be the best candidate. This work guides a novel direction on the effect of other small groups on the EDL, such as -OH, -SO₃, -CO₃, -NH₂, -X (F, Cl, Br, I), etc. We will continue further research, hoping this idea can inspire more researchers.

## Methods

### First-principle calculations
The calculations in this work were performed using spin-polarized density functional theory (DFT) as implemented in the Vienna Ab initio

Simulation Package (VASP). Projector-augmented wave (PAW) potentials with the generalized gradient approximation (GGA) of Perdew-Burke-Ernzerhof (PBE) were used as the exchange-correlation potential. The optimization thresholds were $10^{-5}$ eV and 0.01 eV/Å for electronic and ionic relaxations, respectively. Grimme's DFT-D3 methodology was used to describe the dispersion interactions. A vacuum region of 15 Å was adopted to avoid the interaction of periodic images. During structural optimizations and self-consistent field calculations for the surface models, a $3 \times 3 \times 1$ Gamma-centered Monkhorst-Pack k-point mesh for Brillouin zone sampling was used and all atoms were allowed to relax.

The work function of a material can be calculated by:

$$\Phi = E_{\text{Vac}} - E_{\text{F}} \tag{2}$$

where $E_{\text{Vac}}$ and $E_{\text{F}}$ were the calculated Vacuum energy and Fermi energy, respectively.

The adsorption energies ($E_{\text{ads}}$) of a hydrogen atom on the different substrates were calculated using:

$$E_{\text{ads}} = E_0 - 1/2 E_{\text{H2}} - E_{\text{s}} \tag{3}$$

where $E_0$ and $E_{\text{s}}$ were the total energies per cell with and without the adsorbed hydrogen atom, respectively, and $E_{\text{H2}}$ was the total energy of a hydrogen molecule.

The hydrogen adsorption free energy was calculated at zero potential and pH = 0 as:

$$\triangle G_H = \triangle E_{ads} + \triangle E_{ZPE} - T \triangle S \tag{4}$$

$\Delta G_H$ was then calculated by correcting for both the zero-point vibrational energy and the loss of translation entropy of $H_2(g)$ on adsorption, and neglecting the smaller vibrational entropy terms. The zero-point contributions are essentially identical, yielding $\Delta G_H = \Delta E_{ads} + 0.3$ eV.

## Materials synthesis

Unless otherwise stated, the purities of all reagents for photocatalyst preparation and for photoelectrochemical measurements are above the analytical grade.

**Preparation of CuNi complex compound.** Dissolved 0.003 mol $Cu(NO_3)_2 \cdot 3H_2O$, 0.003 mol $Ni(NO_3)_2 \cdot 6H_2O$, and tartaric acid in a certain amount of deionized water to obtain solution A; Dissolved 0.0005 mol of polyethylene glycol 10000 in deionized water, stirred and sonicated repeatedly to completely dissolve them to obtain solution B; Under continuous stirring, added solution A to solution B drop by drop, and stirred until a uniform and the clear mixture was obtained; Transferred the above mixture to a 100 ml polytetrafluoroethylene reactor, raised the temperature from room temperature to 180°C and kept for 2 h, then cooled it naturally to room temperature. The obtained product was centrifuged and washed with deionized water and ethanol for at least three times until the supernatant was transparent. The precipitate was collected, dried, and fully ground to obtain CuNi complex compound. For the synthesis of CuNi(1:2) and CuNi(2:1), it is basically the same as that of CuNi, except that the additional amounts of Cu source and Ni source in the precursor are different, which are 0.002 mol: 0.004 mol and 0.004 mol: 0.002 mol, respectively.

**Preparation of CuNi@O-C=O.** The dried CuNi complex powder was calcined in a tubular furnace in Ar atmosphere, where it was uniformly raised to 500°C at a heating rate of 5°C/min from room temperature, and the reaction is held for 3 h. After it cooled to room temperature,

taken out and ground with a mortar to obtain CuNi@O-C=O (CN@EDL).

**Preparation of pinnate structured CdS.** Added 0.005 mol of $Cd(CH_3COO)_2 \cdot 2H_2O$ and 0.005 mol of $NH_2CSNH_2$ into 80 ml of water and stirred continuously. After dissolution, added a drop of hydrofluoric acid under stirring. Transferred the above suspension to a 100 ml polytetrafluoroethylene reactor, heated to 200°C, and kept for 30 h. The obtained product was centrifuged and washed. The precipitate was collected, dried, and fully ground to obtain CdS.

**Preparation of CN@EDL/CdS.** Dispersed the obtained CN@EDL in 40 ml deionized water according to the mass percentage of 2%, 5%, and 8%, repeatedly sonicated and stirred. Then 0.0025 mol $Cd(CH_3COO)_2 \cdot 3H_2O$, 0.0025 mol thiourea, a drop of hydrofluoric acid was added to the suspension and stirred for 5 h. Transferred the above suspension to the polytetrafluoroethylene reactor, heated it from room temperature to 200°C in a drying oven for 30 h, and then cooled it naturally to room temperature. The obtained product was centrifuged and washed. The precipitate was collected, dried and fully ground to obtain CuNi@O-C=O/CdS.

**Preparation of Cu@O-C=O and Ni@O-C=O.** The preparation method of Cu@O-C=O (Cu@EDL) and Ni@O-C=O (Ni@EDL) is similar to that of CN@EDL, except that the precursor is $Cu(NO_3)_2 \cdot 3H_2O$ or $Ni(NO_3)_2 \cdot 6H_2O$ instead of both $Cu(NO_3)_2 \cdot 3H_2O$ and $Ni(NO_3)_2 \cdot 6H_2O$.

**Preparation of CuNi.** 0.001 mol of $Cu(NO_3)_2 \cdot 3H_2O$ and 0.01 mol of $Ni(NO_3)_2 \cdot 6H_2O$ were dissolved in 60 mL of 2 M NaOH solution under stirring, followed by adding 0.01 mol sodium hypophosphite into the mixed solution. Finally, transfer the mixture into a 100 mL Teflon-lined autoclave, heated to 140°C and kept for 15 h. The CuNi products were collected by centrifugation and washed by absolute ethanol and distilled water.

**Preparation of CuNi/CdS.** The preparation method of CuNi/CdS is the same as that of CN@EDL/CdS, except that CN@EDL was replaced by CuNi.

## Material characterization

The crystalline phase of the samples was studied by X-ray diffraction (XRD, Bruker, D8 Discover) measurements with Ni-filtered Cu Kα radiation (40 kV, 40 mA). The morphologies of the samples were displayed by scanning electron microscopy (SEM, Hitachi S-4800, Japan) images. The transmission electron microscopy (TEM) images, high-resolution transmission electron microscopy (HRTEM) images, scanning transmission electron microscopy (STEM) images and energy dispersive spectrometer (EDS) elements mapping of the sample were collected by the transmission electron microscopy (TEM, Tecnai G2 F20 S-TWIN, FEI). X-ray photoelectron spectroscopy (XPS) measurements were conducted on a Kratos XSAM-800 spectrometer. Determination of the metal elements content was measured by ICP-OES (Inductively Coupled Plasma Optical Emission Spectrometer, Perkin-Elmer, USA). Fourier transform-infrared (FT-IR) spectra were measured using a VERTEX 70 infrared spectrometer in the region of 4000 - 400 cm$^{-1}$ with a resolution of 2 cm$^{-1}$ based on the CRCP-7070-A in situ transient analysis platform. Raman spectra were carried out on Renishaw-invia RAMAN microscope in the range of 4000 - 40 cm$^{-1}$. Ultraviolet-visible diffuse reflectance spectroscopy (UV-Vis DRS) characterization was performed on the SHIMADZU UV-3600iPLUS spectrophotometer equipped with an integrating sphere, using 100% $BaSO_4$ as the reflectance standard. The static photoluminescence spectra (PL) and time-resolved photoluminescence decay (TRPL) were recorded on the HORIBA FluoroMax-4. A transient population of the carrier was impulsively excited in the sample (maintained in a dark

environment) using an ultraviolet light-emitting diode source ($\lambda = 390$ nm). The spectroscope was coupled with a time-correlated single photon counting module at the 390 nm light-emitting diode excitation at room temperature. To analyze the time-resolved traces, a multiexponential function was used to fit the photoluminescence decay spectra. In our measurements, a biexponential decay function was more suitable based on the smaller value of chi-square. The standard deviations of lifetime were calculated based on three independent samples.

The band gaps of CdS and CuNi@O-C=O/CdS are obtained according to the Kubelka–Munk equation (Eq. 5):

$$\alpha h\upsilon = A\left(h\upsilon - E_g\right)^{1/2} \tag{5}$$

where $\alpha$ is the absorption coefficient, h$\upsilon$ represents the photon energy (h is Planck constant and $\upsilon$ is the frequency of light), $E_g$ is the energy band gap value.

The flat-band potential and carrier density can be obtained according to Eq. 6 and Eq. 7:

$$1/C^2 = 2(E - E_{FB} - k_B T/e)/\varepsilon\varepsilon_0 eN_D \tag{6}$$

$$N_D = (2/e\varepsilon\varepsilon_0)/[d(1/C^2)/dV] \tag{7}$$

The meanings and values of parameters in the above formulas are shown in Supplementary Table 8.

The relationship between $E_{CB}$, $E_{VB}$ and $E_g$ is shown in Eq. 8:

$$E_{VB} = E_{CB} + E_g \tag{8}$$

where $E_{VB}$ is the valence band potential, $E_{CB}$ is the conduction band potential, $E_g$ is the energy band gap value.

## Photocatalytic H₂ evolution measurements

In order to study the hydrogen evolution activity and stability of the designed photocatalyst, the experiment of photocatalytic hydrogen production by water splitting was carried out (Perfect Light Labsolar-6A high air tightness photocatalytic analysis system). Specifically, dispersing 20 mg sample powder into 100 ml aqueous solution (10 ml lactic acid and 90 ml deionized water). Equipping with a 300 W Xenon lamp (Perfect Light PLS-SXE300) as the light source, and using the optical filter ($\lambda > 420$ nm) to obtain the visible light. Before starting reaction, fill the hydrogen production equipment with high-purity nitrogen (N₂) for half an hour to remove the air in the reaction unit. The temperature of the system was maintained at 10°C using a low-temperature thermostat bath (DC-0506). The hydrogen production equipment is connected with GC9790II gas chromatograph to detect the amount of hydrogen. To achieve the apparent quantum efficiency (AQE), band-pass filter (PLS-BP20420, FWHM = 21 nm) was employed under a 300 W Xe lamp (Perfect Light PLS-SXE300). For the measurement of STH efficiency, the standard AM1.5 G filter was employed under a 300 W Xe lamp (Perfect Light PLS-SXE300). Use pl-ms-2000 optical power meter to measure the optical power density (P).

For 0.5 wt%, 1 wt%, 3 wt%, 5 wt% Pt/CdS, we used the in situ photo-deposition method. First, CdS was dispersed in a mixed solution of water and lactic acid, and then a certain amount of 0.003 g mL⁻¹ chloroplatinic acid (H₂PtCl₆·6H₂O) solution was added with a pipette, and irradiated with a full-spectrum Xenon lamp for half an hour. Hydrogen production test method is the same as above.

The AQE can be calculated by Eq. 9, and the meaning and value of each parameter is shown in Supplementary Table 9.

$$AQE = \frac{number\ of\ reacted\ electrons}{number\ of\ incident\ photons} \times 100\% = \frac{2 \times n(H_2) \times N_A}{PSt\lambda/hc} \times 100\% \tag{9}$$

The STH efficiency can be calculated by Eq. 10:

$$STH = \frac{R(H_2) \times \triangle Gr}{P_{sun} \times S} \times 100\% \tag{10}$$

Where $R(H_2)$ is the photocatalytic hydrogen evolution rate (20 mg, mmol s⁻¹), $\triangle Gr$ is the molar Gibbs free energy of water splitting reaction (237 kJ mol⁻¹), $P_{sun}$ is the optical power density of the AM1.5 G standard solar spectrum (100 mW cm⁻²), $S$ is the irradiation area (19.6 cm²), the STH efficiency is calculated to be 3.83%.

## Electrochemical measurements

The electrochemical measurements were carried out in a standard three-electrode cell system with a Fluorine doped tin oxide (FTO) deposited with catalysts as the working electrode, a Ag/AgCl (KCl saturated) electrode as the reference electrode and a carbon electrode as the counter electrode. The working electrode was fabricated as follows: Mix the catalyst with PVDF (polyvinylidene difluoride) according to the mass ratio of 6:1, add two drops of DMF (N,N-Dimethylformamide) to form a homogenate, then disperse it evenly by ultrasound. Afterwards, drop-cast it to FTO glass with a coating area of 1 cm × 1 cm. The light for the photoelectrochemical measurements was the filtered light ($\lambda > 420$ nm) from a PLS-SXE300 Xenon lamp (Perfect Light, Beijing). The Mott-Schottky curves (M-S), electrochemical impedance spectroscopy (EIS), and linear sweep voltammetry (LSV) measurements were performed on the CS350H electrochemical workstation. The EIS was measured at a 5 mV alternating current signal in the frequency range of 0.01–10⁶ Hz. A 0.5 M Na₂SO₄ aqueous solution was used as the electrolyte of M-S and EIS tests. For LSV measurements, 0.5 M Na₂SO₄, 0.5 M H₂SO₄, and 1.0 M KOH aqueous solution were used as electrolytes, respectively.

## Data availability

All data supporting the findings of this study are available within the article, as well as the Supplementary Information file, or available from the corresponding authors.

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

## Author contributions

C.Z. conceived the project and conducted the experiments, drew figures, and analyzed the calculation and experimental data. J.G. and C.X. gave advice on the experiments. D.L., Y.D., M.W., and C.X. participated in the grammatical revision of the paper. C.Z. wrote the manuscript. All authors discussed the results and contributed to the manuscript.

## Competing interests

The authors declare no competing interests.
