## [Peer Review File · Nature Communications]

REVIEWER COMMENTS

Reviewer #1 (Remarks to the Author):

The article describes a fundamentally new method for the synthesis of photocatalysts based on the construction of an electric double layer. By this method, the authors obtained photocatalysts based on cadmium sulfide with an activity much higher than the data reported in the literature. The article is well structured, the photocatalysts are described by a complex of modern characterization methods, and the experimental data do not raise doubts about their reliability. During the study of the article, only minor remarks arose.

1. Value 249.6 mmol h⁻¹ g⁻¹ - too many significant digits.
2. Element mapping (Fig. 3) has very poor resolution; the scale of 100 nm and 1 μm is too large. These figures do not provide any reliable information.
3. Fig. 5a. The error of measurements should be shown.
4. Fig 5i. The figure with the scheme of the composite photocatalyst and the hydrogen evolution mechanism is too small and indistinct. The resulting heterojunctions, etc., should be clearly shown.

Reviewer #2 (Remarks to the Author):

In this manuscript, Zhou et al. reported the synthesis of ester modified CuNi HER cocatalyst, and the photocatalytic performance of CN@EDL/CdS was investigated. This work is interesting and has done quite a detailed study. This manuscript can be considered the publication in Nature Communications after revision. The detailed comments are given below.

1. How did the control sample CuNi/CdS prepared? The Cu and Ni content of this sample should be analyzed by ICP.
2. In page 8, line 147, I'm confused with the sentence "The signals of CN@EDL are weak due to the low content and high dispersion." It seems the intensity of CN@EDL is not weak, as shown in Fig. 3a.
3. Why does the CdS used in the MS have a snowflake morphology instead of other morphology, such as block and nanosheet? Does shape affect catalytic activity?
4. Why choose CdS, NOT other semiconductors? Such as TiO₂, Ta₃N₅, carbon nitride? To demonstrate the universality of CN@EDL, other supports should also be investigated.

5. As shown in Fig. 4d, 4e and Supplementary Figure 18 (b and c), these XPS curves look very smooth without any baseline noise, why?
6. The author gave the characterization of samples (CN@EDL/CdS) with different Cu: Ni ratios, but did not give their catalytic activities. For this part, the author should discuss and analyze the structure-activity of CN@EDL/CdS with different Cu:Ni ratios.
7. When comparing the difference in the catalytic activities between CN@EDL/CdS and Pt/CdS, the same content of co-catalysts should be used, or the one with the highest activity should be compared, and it should be clearly marked in the main text. Is 1% optimal for Pt as a co-catalyst? What about 5%?
8. The authors should examine the effect of different sacrificial agents on the PHE activity, such as TEOA, MeOH, Na₂S/Na₂SO₃, etc.
9. I think there is a problem with the expression of CN@EDL (x%). Does x% means the total content of Cu and Ni? What are the contents of Cu and Zn respectively? These samples should be analyzed by ICP and the results are suggested to present in tabular form.
10. The author should investigate the AQE of CN@EDL/CdS at different wavelengths.
11. In order to prove the stability of CN@EDL/CdS, the author should give the XPS results of the sample after 300 days to prove whether Cu and Ni are oxidized?
12. How about the PHE activities of Cu@O-C=O/CdS and Ni@O-C=O/CdS?
13. In the experimental section, what does "CuNi complex compound" mean? If the CuNi complex has a fixed composition, the author should give the elemental analysis and ICP analysis results.
14. TEM study and ICP analysis (for Cu and Ni) of the catalyst after a long-time scale HER are preferred.

Reviewer #3 (Remarks to the Author):

The dynamics and thermodynamic performance of carrier are important for photocatalytic H₂ evolution. Xia. et. al. proposed to introduce electronegative groups into the cocatalyst to build the electric double layer and improved the carrier dynamics and thermodynamics. The designed cocatalyst CuNi@EDL improved the H₂ evolution rate to 249.6 mmol h⁻¹ g⁻¹, which outperforms the reported catalysts. But there are many concerns which need to be addressed. Hence I withhold my decision for now, after looking at the revision, the decision can be made.

1. Lactic acid is used as sacrificial agent in the photocatalysis. Is lactic acid acting as hydrogen source here? The isotope experiments are recommended to definite the hydrogen source. Photocatalysis with different sacrificial agents should also be conducted.
2. The authors claim that the catalysts remained stable after more than 300 days. Is there any photocatalytic experiments data for 300 days? Figure S31 only shows the XRD of the CN@EDL(5%)/CdS stored for 300 days, instead of after photocatalysis for 300 days.
3. The authors described that “The superior yield of H₂ is mainly due to the perfect work function, Fermi level and Gibbs free energy of hydrogen adsorption, improved light absorption ability, enhanced electron transfer dynamics, decreased HER overpotential and effective carrier transfer channel arose by EDL”. What’s the relationships between the above factors? Which factor is the most important one?
4. More detailed descriptions should be provided for the preparation of CuNi@O-C=O and CuNi@C=O. What’s the inert atmosphere in a tubular furnace?
5. Can the loading amount of carbonyl or ester groups on CuNi be controlled precisely? What’s the loading amount of the groups in the cocatalysts used in this work? More characterizations should be provided here.
6. Work functions of other cocatalysts, such as CuNi@O-C=O should be added in Figure S2.
7. The intensities differences in light absorption and steady-state PL spectra are not convincing as the samples amount may be different.

Reviewer #4 (Remarks to the Author):

Comments to NCOMMS-22-47760

The authors describe a novel strategy to boost the photocatalytic performance of the catalysts by coating the CuNi cocatalyst with electronegative molecules. The coating introduces an EDL, which is responsible for the enhanced photocatalytic performance of the catalysts. However, the DFT calculations in the manuscript are quite simplified and can not sufficiently support the idea. And the experimental evidence is vague and lacks a determining proof. Furthermore, the figures in the SI are not quite high in quality. Some critical concerns should be addressed before its publication in this or other journals. Based on the facts mentioned above, I do not suggest its publication in Nature Communications.

Q1: The models used in this research are quite simplified and can not support the core idea of the authors. The lengths of the model in the x and y directions are quite small and cannot eliminate the interactions from the adjacent supercell, especially those between the functional groups. Due to the quantum description of the functional groups, could the authors discuss the difference between the ester group and the carbon dioxide molecules? From my point of view, the functional group that is put on the surface of the CuNi is quite the same as the CO₂ molecules. So, would the CO₂ or CO molecules work similarly and can generate the EDL? Moreover, the models in Figure S1 indicate the presence of two functional groups, why? Would they influence each other? A caption indicating the species of the atoms should be added in Figure S1. The distance between the carbon atoms that bond to the metal atoms should be provided to determine whether a chemical bond forms. Furthermore, different configurations of the functional groups adsorbed on the surfaces should be compared to find the optimum one. Overall, the DFT models should be carefully checked and modified to satisfy these requirements.

Q2: The authors proposed a possible working principle of the EDL. However, limited evidence is provided to support this mechanism. The authors are suggested to offer both experimental and theoretical evidence to demonstrate the possible principles.

Q3: The authors are suggested to cite the classic work of Nørskov et al. to honor their contribution to the d-band theory.

Q4: No citations can be found except in the Introduction section. The authors are advised to check the manuscript carefully and provide the necessary some.

Q5: Via a simple hydrothermal method, the CN@EDL can be obtained. So where does the O-C=O come from? A detailed investigation of the formation process of this unique structure should be carried out. What is the exact form of the O-C=O? As a molecule or as a free radical? A schematic illustration would be fine.

Q6: How the EDL promotes photocatalytic activity? A detailed discussion is needed to clarify the role it plays. What is the STH efficiency for this unique structure? A comparison with the recently reported ones in terms of the STH efficacy and activity should be provided to prove its advantage.

Q7: The authors claim that the catalysts “remained stable after more than 300 days”. Does the photocatalytic activity remain stable? If so, prove it.

Q8: What do the symbols in the equations in the Methods mean? The authors are strongly suggested to provide the information.

Q9: The authors are advised to pay attention to the aesthetics of the Figures. The font of the text and the length of the scale bar in Figure 3 is complete chaos. The discrepancy in font size, font, and position of the caption in the Figures in the SI are quite common. The author should pay more attention to these problems. Furthermore, the authors are strongly suggested polishing their manuscript to avoid some possible typos or grammatical mistakes.

Response letter to reviewer 1

Dear reviewer,

Thank you very much for your meaningful suggestions, we have learned a lot from your points. According to your comments, we have made detailed modifications to the manuscript and responded to each question. The specific contents are as follows:

[The article describes a fundamentally new method for the synthesis of photocatalysts based on the construction of an electric double layer. By this method, the authors obtained photocatalysts based on cadmium sulfide with an activity much higher than the data reported in the literature. The article is well structured, the photocatalysts are described by a complex of modern characterization methods, and the experimental data do not raise doubts about their reliability. During the study of the article, only minor remarks arose.]

Response: Thank you very much for your positive affirmation of our article. We have learned a lot from your comments.

1. Value $249.6 \text{ mmol h}^{-1} \text{ g}^{-1}$ - too many significant digits.

Response: Thank you very much for your comments. In order to eliminate accidental errors, we use the average value of multiple tests and reserve one decimal point to make it more accurate. If you think it is inappropriate, we will make further modifications according to your opinions.

2. Element mapping (Fig. 3) has very poor resolution; the scale of 100 nm and 1 μm is too large. These figures do not provide any reliable information.

Response: Thank you very much for your comments. We have replaced Fig. 3 with the original image to improve the resolution as shown in the revised manuscript (Fig. 3). For the problem that the scale of 100 nm and 1 μm is too large, we consider that the scale of the substrate material CdS is up to several micrometers (μm) and CN@EDL nanoparticles with a size of tens of nanometers (nm), therefore, the scale of the element mapping is 100 nm and 1 μm respectively. The above is only our consideration of this question, if there are something inappropriate, please do not hesitate to contact us.

3. Fig. 5a. The error of measurements should be shown.

Response: Thank you very much for pointing out our shortcomings. We have supplemented the error bar of hydrogen production measurements, as shown in **Figure 5a** in the revised manuscript (the standard error values in Figure 5a are calculated to be 0.6553, 0.9392, 4.6530, 2.2798, respectively). And according to your suggestion, we have added the error bars in other figures, such as **Supplementary Figure 22-24**. Error bars are standard error values of three tests ($n=3$). Example calculation formula: $A4=STDEV.S(A1:A3)$, $A5=A4/SQRT(3)$ (using Excel tables).

4. Fig 5i. The figure with the scheme of the composite photocatalyst and the hydrogen evolution mechanism is too small and indistinct. The resulting heterojunctions, etc., should be clearly shown.

Response: Thank you very much for your comments. We have removed the background in the schematic diagram in Figure 5i and enlarged the font to make it clearer, as shown in the revised manuscript.

Fig. 5 | Photocatalytic activity and mechanism research. h, LSPR effect in CN@EDL and h - e^- generation progress. **i,** Energy band diagram of CN@EDL/CdS.

The above is our response to the comments, and the specific modifications are shown in the revised manuscript and SI file. Please don't hesitate to contact us if you have any comment about our revised manuscript. We are looking forward to receiving your valuable comments to make the manuscript more perfect. And thank you once more for your valuable comments!

Sincerely,

Chengxin Zhou

Response letter to reviewer 2

Dear reviewer,

Thank you very much for your valuable comments, we have learned a lot from your suggestions. According to your comments, we have made detailed modifications to the manuscript and made responses to each question. The specific contents are as follows:

[In this manuscript, Zhou et al. reported the synthesis of ester modified CuNi HER cocatalyst, and the photocatalytic performance of CN@EDL/CdS was investigated. This work is interesting and has done quite a detailed study. This manuscript can be considered the publication in Nature Communications after revision. The detailed comments are given below.]

Response: Thank you very much for your positive comments on our work. We have learned a lot from your comments.

1. How did the control sample CuNi/CdS prepared? The Cu and Ni content of this sample should be analyzed by ICP.

Response: Thank you for your comments. The preparation method of CuNi is as described in '*Preparation of CuNi*'. The preparation method of CuNi/CdS is the same as that of CN@EDL/CdS, except that CN@EDL was replaced by CuNi. According to your comment, we have added the preparation method of CuNi/CdS, as shown in the revised manuscript.

Preparation of CuNi. 0.001 mol of $\text{Cu}(\text{NO}_3)_2 \cdot 3\text{H}_2\text{O}$ and 0.01 mol of $\text{Ni}(\text{NO}_3)_2 \cdot 6\text{H}_2\text{O}$

were dissolved in 60 mL of 2 M NaOH solution under stirring, followed by adding 0.01 mol sodium hypophosphite into the mixed solution. Finally, transfer the mixture into a 100 mL Teflon-lined autoclave, heated to 140°C and kept for 15 h. The CuNi products were collected by centrifugation and washed by absolute ethanol and distilled water.

Preparation of CuNi/CdS. The preparation method of CuNi/CdS is the same as that of CN@EDL/CdS, except that CN@EDL was replaced by CuNi.

The content of Cu and Ni in the control cocatalyst CuNi was measured to be 50.66 wt% (3.510 mg/L) and 49.34 wt% (3.157 mg/L) by ICP respectively, as shown in the revised SI file (**Supplementary Table 10 and 11, SI file Page 50 and 51**).

Supplementary Table 10. The CuNi contents in the as-prepared samples measured by ICP-OES.

Samples	Cu (mg/L)	Cu (wt%)	Ni (mg/L)	Ni (wt%)	CuNi (wt%)
CN@EDL(2%)/CdS	0.295	1.52	0.292	1.50	3.02
CN@EDL(5%)/CdS	0.361	1.93	0.474	2.54	4.47
CN@EDL(8%)/CdS	0.449	3.49	0.521	4.02	7.51
CuNi(5%)/CdS	0.361	1.94	0.474	2.53	4.47
CN@EDL(5%)/CdS Used sample	0.334	1.93	0.437	2.43	4.36

Supplementary Table 11. The Cu and Ni content in CuNi(1:2, 1:1 and 2:1) measured by ICP-OES.

Samples	Cu (mg/L)	Cu (atom ratio%)	Ni (mg/L)	Ni (atom ratio%)
CuNi(1:2)@EDL	2.500	30.2	5.375	69.8
CuNi(1:1)@EDL	1.260	46.5	1.385	53.6
CuNi(2:1)@EDL	6.876	70.1	2.709	29.9
CuNi(1:1)	3.510	50.7	3.157	49.3

2. In page 8, line 147, I'm confused with the sentence "The signals of CN@EDL are weak due to the low content and high dispersion." It seems the intensity of CN@EDL is not weak, as shown in Fig. 3a.

Response: Thank you very much for your comments. We originally considered that the content of CN@EDL was much lower in comparison with that of CdS and the diffraction peak was relatively weaker, so we expressed it as "The signals of CN@EDL are weak due to the low content and high dispersion". As you pointed out, the diffraction peak of CN@EDL is not particularly weak, but relatively weaker than that of CdS, so we deleted this sentence according to your opinion, as shown in the revised manuscript.

3. Why does the CdS used in the MS have a snowflake morphology instead of other morphology, such as block and nanosheet? Does shape affect catalytic activity?

Response: Thank you very much for your comments. In the field of photocatalysis, many excellent studies have shown that the size and morphology of the catalyst will

affect its performance. In our previous studies, we also found that CdS with different morphologies showed different hydrogen production activities under the same conditions (it has a certain impact, but not a special impact). Considering the photosynthesis of leaves in nature, we designed a snowflake CdS symmetrically assembled by leaves (because CdS belongs to hexagonal crystal system, it is self-assembled by leaves into hexagonal snowflakes). However, considering that the core idea of this study is to design a kind of cocatalyst, which can effectively improve the hydrogen production activity by constructing an electric double layer. Therefore, the type and morphology of the main photocatalyst is not the focus of this study, and due to the limitation of manuscript length, we only describe the key content in this article. Thank you once more for your suggestion and we will take shape factor into account in the further study.

4. Why choose CdS, NOT other semiconductors? Such as TiO₂, Ta₃N₅, carbon nitride?
To demonstrate the universality of CN@EDL, other supports should also be investigated.

Response: Thank you very much for your comments. At the beginning of the study, we chose CdS as the main photocatalyst after comprehensive consideration, mainly due to the appropriate band gap and energy level positions of CdS ($E_g=2.44$ eV, $CB=-0.68$ V, $VB=1.76$ V) for photocatalytic hydrogen production. As for TiO₂, the band gap is too large (3.2 eV) to effectively utilize the visible light. As for carbon nitride, its band gap and energy level positions are relatively appropriate, however, the conductivity of non-

metallic semiconductors is not good, the recombination of photogenerated electron-hole is serious, the light absorption efficiency is low, the photocatalytic performance of carbon nitride is not ideal. As for Ta₃N₅, it has a narrow band gap ($E_g=2.1$ eV) and can make full use of the visible light. However, Ta is a rare metal with high price, which increases the cost of hydrogen production. Meanwhile, its practical application in the field of photocatalysis is limited by its disadvantages such as poor carrier transport ability, easy recombination of carriers, low separation efficiency and poor stability due to photocorrosion. Moreover, the synthesis of CdS is relatively simple and does not require high temperature calcination and other energy-consuming conditions, therefore, we chose CdS as the photocatalyst. The above is our response to this question, if there is anything inappropriate, please do not hesitate to contact us.

5. As shown in Fig. 4d, 4e and Supplementary Figure 18 (b and c), these XPS curves look very smooth without any baseline noise, why?

Response: Thank you very much for your comments. In order to show the valence states of the target elements more clearly, we have carried out smooth-fit on some XPS curves. However, it does not affect the intensity of characteristic peaks of elements and the corresponding binding energy positions. As you pointed out, the XPS curve of some elements with low content has noise, so we have taken the above treatment to make its expression clearer. If this is inappropriate, we will make further modifications.

6. The author gave the characterization of samples (CN@EDL/CdS) with different Cu:

Ni ratios, but did not give their catalytic activities. For this part, the author should discuss and analyze the structure-activity of CN@EDL/CdS with different Cu:Ni ratios.

Response: Thank you very much for your comments. We think this is a very meaningful question because the element ratio may play an important role in the properties of the alloy. Therefore, we originally synthesized CuNi(1:2)@EDL, CuNi(1:1)@EDL and CuNi(2:1)@EDL. However, we found that CuNi(1:2) and CuNi(2:1) were not fully alloyed from the XRD analysis (**Supplementary Figure 10**), so we considered that the activity of CuNi(1:2) and CuNi(2:1) may be lower than that of CuNi(1:1). What's more, we considered that the focus of this study is on the role of electric double layer, therefore, the influence of the ratio of Cu to Ni on the performance is not fully analyzed. According to your suggestion, we have supplemented the hydrogen production tests of CuNi(1:2)@EDL/CdS and CuNi(2:1)@EDL/CdS (each sample has been tested at least three times), the results are as shown in the revised SI file (**Supplementary Figure 22**). The analysis found that compared with single CdS, the hydrogen production activities of CuNi(1:2)@EDL/CdS and CuNi(2:1)@EDL/CdS were both improved, and CuNi(1:2)@EDL/CdS was more obvious, but both were far lower than CuNi(1:1)@EDL/CdS. It shows that the atom ratio does affect the photocatalytic activity of CuNi@EDL and CuNi(1:1)@EDL is more conducive to photocatalytic hydrogen production.

Supplementary Figure 22 Hydrogen production rates of (a) Cu@EDL/CdS, (b) Ni@EDL/CdS, (c) CN(1:2)@EDL/CdS and (d) CN(2:1)@EDL/CdS. Error bars are standard error values of three tests (n=3).

7. When comparing the difference in the catalytic activities between CN@EDL/CdS and Pt/CdS, the same content of co-catalysts should be used, or the one with the highest activity should be compared, and it should be clearly marked in the main text. Is 1% optimal for Pt as a co-catalyst? What about 5%?

Response: Thank you very much for your comments. As you pointed out, the loading amount of Pt does affect the photocatalytic hydrogen production activity. It is our negligence that the hydrogen production performance of Pt/CdS with various Pt loading amounts. According to your suggestion, we have supplemented the hydrogen production activity of samples with different amounts of Pt, such as 0.5wt%, 1wt%, 3wt%, 5wt%. For 0.5wt%, 1wt%, 3wt%, 5wt% Pt/CdS, 19.9 mg, 19.8 mg, 19.4 mg, 19.0 mg of CdS powder were dispersed into 100 ml aqueous solution (10 ml lactic acid and 90 ml deionized water) respectively, then 88.5 μL, 177.1 μL, 531.3 μL, 885.5 μL

of 0.003 g mL⁻¹ chloroplatinic acid (H₂PtCl₆·6H₂O) solution was added with a pipette respectively, and irradiated with a full-spectrum Xenon lamp for half an hour.

The results are shown in the revised SI file (**Supplementary Figure 23**), and it is found that the hydrogen production activity reached the highest when the content of Pt is 1wt%. There is an optimal value for the loading amount of the cocatalyst, and the performance will decline if it exceeds the optimal value owing to the agglomeration of Pt and a reduced utilization efficiency of Pt.

Supplementary Figure 23 Hydrogen production rates of Pt/CdS with different Pt loading amounts (0.5wt%, 1wt%, 3wt% and 5wt%). Error bars are standard error values of three tests (n=3).

8. The authors should examine the effect of different sacrificial agents on the PHE activity, such as TEOA, MeOH, Na₂S/Na₂SO₃, etc.

Response: Thank you very much for your comments. According to your suggestion, we have tested the hydrogen production performance with different sacrificial agents such as 10vol% TEOA, 10vol% MeOH and 0.25 M Na₂S/0.35 M Na₂SO₃ respectively

and found that the hydrogen production rates follow the order of 10vol% lactic acid > 0.25 M Na₂S / 0.35 M Na₂SO₃ > 10vol% TEOA > 10vol% MeOH. It can be found that the type of sacrificial agent does affect the hydrogen production activity, possibly because different sacrificial agents correspond to different oxidation potentials, and the easier it is to be oxidized by holes, the faster the hole consumption is, and the higher the hydrogen production rate is. However, according to the literature, different studies found that different sacrificial agents are better, which may be due to different sacrificial agents are suitable for different photocatalyst systems.

9. I think there is a problem with the expression of CN@EDL (x%). Does x% means the total content of Cu and Ni? What are the contents of Cu and Ni respectively? These samples should be analyzed by ICP and the results are suggested to present in tabular form.

Response: Thank you very much for your comments. The expression of x% in CN@EDL/CdS means the total content of CN@EDL in the composite. We found that when the feed ratio of Cu and Ni is 1:1, it is more favorable for alloying and has higher hydrogen production activity. Therefore, we focus on the performance of CuNi(1:1)@EDL. Here, x% represents the percentage of feed mass of CuNi(1:1)@EDL in CuNi(1:1)@EDL/CdS. As you pointed out, it is important to analyze the contents of Cu and Ni respectively, therefore, we tested the contents of Cu and Ni by ICP according to your suggestion, as shown in the revised SI file (**Supplementary Table 10** and **11**, or Response letter **Page 5-6**).

10. The author should investigate the AQE of CN@EDL/CdS at different wavelengths.

Response: Thank you very much for your comments. We initially considered that the main research of this study is visible light photocatalytic hydrogen production, so we focused on testing and calculating the AQE at 420 nm. According to your suggestion, we have investigated the AQE of CN@EDL/CdS at 365 nm, 420 nm and 520 nm, respectively. Band-pass filterS (PLS-BP20365, FWHM=25 nm, PLS-BP20420, FWHM=21 nm, PLS-BP20520, FWHM=20 nm) were employed under a 300W Xe lamp (Perfect Light PLS-SXE300). We have tested the hydrogen production rates at 365 nm (20 mg, 0.81 mmol h⁻¹) and 520 nm monochromatic light (20 mg, 0.383 mmol h⁻¹), but the optical power density cannot be measured due to a problem with the optical power meter, that is, that is, P in **Equation 5** cannot obtain an accurate value at present. Therefore, based on the scientific attitude, we have not listed AQE at $\lambda=365$ nm and $\lambda=520$ nm. Moreover, the amounts of catalyst used in different researches are different, so the calculated AQE are also different. For example, the more catalyst used under the same conditions, the larger $n(H_2)$, and the larger AQE. In this study, the amount of catalyst used for AQE is 20 mg, and $n(H_2)$ is the amount of hydrogen production under monochromatic light.

The AQE can be calculated by **Equation 5**, and the meaning and value of each parameter is shown in **Supplementary Table 9**.

$$AQE = \frac{\text{number of reacted electrons}}{\text{number of incident photons}} \times 100\% = \frac{2 \times n(H_2) \times N_A}{PSt\lambda/hc} \times 100\% \quad (5)$$

11. In order to prove the stability of CN@EDL/CdS, the author should give the XPS

results of the sample after 300 days to prove whether Cu and Ni are oxidized?

Response: Thank you very much for your comments. In the original SI file, the XRD of sample stored for 300 days has been tested to prove its stability. According to your suggestion, we have supplemented the high-resolution XPS spectra of CN@EDL/CdS after storing for more than one year, as shown in the revised SI file (**Supplementary Figure 34**). There was no obvious difference between it and the fresh sample, indicating that the photocatalyst was relatively stable.

Supplementary Figure 35 High-resolution XPS patterns of CN@EDL/CdS stored for 300 days.

12. How about the PHE activities of Cu@O-C=O/CdS and Ni@O-C=O/CdS?

Response: Thank you very much for your comments. According to your suggestion, we have tested the PHE activities of Cu@O-C=O/CdS and Ni@O-C=O/CdS, and the

results show that the hydrogen evolution rates of them are 13.6 mmol g⁻¹ h⁻¹ and 82.9 mmol g⁻¹ h⁻¹ respectively, as shown in the revised SI file (**Supplementary Figure 22**, or Response letter **Page 10**).

13. In the experimental section, what does “CuNi complex compound” mean? If the CuNi complex has a fixed composition, the author should give the elemental analysis and ICP analysis results.

Response: Thank you very much for your comments. In this preparation process, the CuNi complex compound was first synthesized by hydrothermal and solvothermal method; secondly, the functional group was obtained by high-temperature calcination due to the decomposition of organic components in the precursor. Therefore, “CuNi complex compound” here refers to the amorphous complex formed by CuNi and organics in the first step of the reaction, that is, some organics are attached to the surface of CuNi. It has no fixed form, but a complex mixture of alloy and organic molecular chain. The above is our response to this question. If there is anything inappropriate, please do not hesitate to contact us.

14. TEM study and ICP analysis (for Cu and Ni) of the catalyst after a long-time scale HER are preferred.

Response: Thank you very much for your comments. According to your opinion, we have supplemented the morphology characterization and element analysis of the sample after a long-time reaction, as shown in the revised SI file (**Supplementary Figure 32**

and **Supplementary Table 10**). As shown in **Supplementary Figure 32**, after a long time of reaction and ultrasonic, centrifugal, washing and other processes, the morphology of CN@EDL/CdS has not changed significantly, and most of them still remain snowflake structure. Of course, it is inevitable that there is a certain degree of mechanical peeling or fracture in the process, so there are also some broken snowflakes. From the ICP-OES test results (**Supplementary Table 10, SI file Page 50**), it can be found that after a long time of reaction, the type and content of elements in CN@EDL/CdS have not changed significantly. Further considering the test error, there is almost no change before and after the reaction.

Supplementary Figure 33 SEM patterns of the used CN@EDL/CdS.

Supplementary Table 10. The CuNi contents in the as-prepared samples measured by ICP-OES.

Samples	Cu (mg/L)	Cu (wt%)	Ni (mg/L)	Ni (wt%)	CuNi (wt%)
CN@EDL(2%)/CdS	0.295	1.52	0.292	1.50	3.02
CN@EDL(5%)/CdS	0.361	1.93	0.474	2.54	4.47

CN@EDL(8%)/CdS	0.449	3.49	0.521	4.02	7.51
CuNi(5%)/CdS	0.361	1.94	0.474	2.53	4.47
CN@EDL(5%)/CdS Used sample	0.334	1.93	0.437	2.43	4.36

The above is our response to the comments, and the specific modifications are shown in the revised manuscript and SI file. Please don't hesitate to contact us if you have any comment about our revised manuscript. We are looking forward to receiving your valuable comments to make the manuscript more perfect. And thank you once more for your valuable comments!

Sincerely,

Chengxin Zhou

Response letter to reviewer 3

Dear Reviewer,

Thank you very much for your meaningful suggestions, we have learned a lot from your points. According to your comments, we have made detailed modifications and made responses to each question. The specific contents are as follows:

[The dynamics and thermodynamic performance of carrier are important for photocatalytic H₂ evolution. Xia. et. al. proposed to introduce electronegative groups into the cocatalyst to build the electric double layer and improved the carrier dynamics and thermodynamics. The designed cocatalyst CuNi@EDL improved the H₂ evolution rate to 249.6 mmol h⁻¹ g⁻¹, which outperforms the reported catalysts. But there are many concerns which need to be addressed. Hence I withhold my decision for now, after looking at the revision, the decision can be made.]

Response: Thank you very much for taking the valuable time to guide our research and put forward valuable suggestions. We learned a lot from your comments.

1. Lactic acid is used as sacrificial agent in the photocatalysis. Is lactic acid acting as hydrogen source here? The isotope experiments are recommended to definite the hydrogen source. Photocatalysis with different sacrificial agents should also be conducted.

Response: Thank you very much for your comments. We think this is a very meaningful issue. At present, the laboratory is unable to complete the isotope tracing experiment. Therefore, according to your suggestions, we have tested the hydrogen production

activity of the sample in pure lactic acid and found that its hydrogen production rate was about 0.07 mmol h^{-1} . It can be seen that the hydrogen of this system basically comes from water. Moreover, all the control trials used the method of controlling variables. All samples were tested with the same amount of lactic acid (10vol%), and the content of sacrificial agent is lower than 20vol% or 30vol% used in some literatures. Furthermore, the photocatalysis with different sacrificial agents were conducted according to your suggestions, such as 10vol% TEOA, 10vol% MeOH and 0.25 M Na₂S/0.35 M Na₂SO₃, respectively. It was found that the hydrogen production rates follow the order of 10vol% lactic acid > 0.25 M Na₂S/0.35 M Na₂SO₃ > 10vol% TEOA > 10vol% MeOH. It can be found that the type of sacrificial agent does affect the hydrogen production activity, possibly because different sacrificial agents correspond to different oxidation potentials, and the easier it is to be oxidized by holes, the faster the hole consumption is, and the higher the hydrogen production rate is. However, according to the literature, different studies found that different sacrificial agents are better, which may be due to different sacrificial agents are suitable for different photocatalyst systems. Many studies have been devoted to the analysis of the impact of different sacrificial agents on the photocatalysis system. Due to the limitation of the focus and length of this study, we just choose a commonly used sacrificial agent. The above is our response to this question. If there is anything inappropriate, please do not hesitate to contact us.

2. The authors claim that the catalysts remained stable after more than 300 days. Is there

any photocatalytic experiments data for 300 days? Figure S31 only shows the XRD of the CN@EDL(5%)/CdS stored for 300 days, instead of after photocatalysis for 300 days.

Response: Thank you very much for your comments. We have tested the XRD and XPS of the photocatalysts stored for more than one year under environmental conditions to prove their stability, that is, without oxidation or photo-corrosion, rather than continuous photocatalytic reaction for 300 days. Due to the limitation of Xenon lamp light source and other factors, it is difficult to carry out the photocatalytic test for 300 consecutive days. Therefore, we tested the stability of the photocatalyst after photocatalytic reaction for dozens of hours (such as **Supplementary Figure 26-32**) and the stability of the samples placed for 300 days under ambient conditions (**Supplementary Figure 33-34**). In order to verify the stability of its hydrogen production activity, we conducted the photocatalytic hydrogen production test on the samples placed for more than one year (note: not continuous reaction for 300 days), and the results showed that its photocatalytic performance did not change obviously.

3. The authors described that “The superior yield of H₂ is mainly due to the perfect work function, Fermi level and Gibbs free energy of hydrogen adsorption, improved light absorption ability, enhanced electron transfer dynamics, decreased HER overpotential and effective carrier transfer channel arose by EDL”. What’s the relationships between the above factors? Which factor is the most important one?

Response: Thank you very much for your comments. We think this is a very meaningful

opinion as it is very important to clarify the mechanism. As we know, the key factors affecting the photocatalytic hydrogen production performance include the kinetic performance of electron in the catalyst; the ability to reduce H^+ after the electrons reach the surface of the catalyst; H adsorption and desorption capacity on the catalyst surface. The light absorption performance determines the energy conversion efficiency, because in the first step, photocatalysts absorb the energy of photons ($h\nu$) to produce photogenerated electron-holes (h^+/e^-); Subsequently, the performance of photogenerated electrons affects the performance of hydrogen production; The lower the Fermi energy level, the greater the work function, the stronger the ability to capture electrons. Together with the carrier transport channel formed by EDL, the transport dynamics of electrons are enhanced; The greater the overpotential of hydrogen evolution, the greater the polarization potential of hydrogen evolution. Only under a more negative potential can hydrogen be released, which increases the difficulty of hydrogen evolution and reduces the hydrogen production. Therefore, reducing the overpotential of hydrogen evolution is very important to improve the hydrogen production activity; In addition to the performance of electrons, the Gibbs free energy of hydrogen adsorption affects the adsorption and desorption of hydrogen (capacity to adsorb H^* and release H_2), and is also a key factor affecting the hydrogen production activity.

4. More detailed descriptions should be provided for the preparation of $CuNi@O-C=O$ and $CuNi@C=O$. What's the inert atmosphere in a tubular furnace?

Response: Thank you very much for your comments. For the preparation of CuNi@O-C=O, we adopted the high-temperature calcination method. Specifically, the CuNi complex compound was calcined in a tubular furnace in Ar atmosphere, and it was uniformly raised from room temperature to 500°C at a heating rate of 5°C/min, and the reaction is held for 3 hours. After it cooled to room temperature, taken out and ground with a mortar to obtain CuNi@O-C=O (CN@EDL). The inert atmosphere in a tubular furnace is Ar. We are very sorry for missing this information in the text. We have added it, as shown in the revised manuscript.

Preparation of CuNi complex compound. Dissolved 0.003 mol $\text{Cu}(\text{NO}_3)_2 \cdot 3\text{H}_2\text{O}$, 0.003 mol $\text{Ni}(\text{NO}_3)_2 \cdot 6\text{H}_2\text{O}$ and tartaric acid in a certain amount of deionized water to obtain solution A; Dissolved 0.0005 mol of polyethylene glycol 10000 in deionized water, stirred and sonicated repeatedly to completely dissolve them to obtain solution B; Under continuous stirring, added solution A to solution B drop by drop, and stirred until a uniform and clear mixture was obtained; Transferred the above mixture to a 100 ml polytetrafluoroethylene reactor, raised the temperature from room temperature to 180°C and kept for 2 h, then cooled it naturally to room temperature. The obtained product was centrifuged and washed with deionized water and ethanol for at least 3 times until the supernatant was transparent. The precipitate was collected, dried and fully ground to obtain CuNi complex compound. For the synthesis of CuNi(1:2) and CuNi(2:1), it is basically the same as that of CuNi, except that the addition amounts of Cu source and Ni source in the precursor are different, which are 0.002 mol : 0.004 mol and 0.004

mol : 0.002 mol, respectively.

Preparation of CuNi@O-C=O. The dried CuNi complex powder was calcined in a tubular furnace in Ar atmosphere, where it was uniformly raised to 500°C at a heating rate of 5°C/min from room temperature, and the reaction is held for 3 hours. After it cooled to room temperature, taken out and ground with a mortar to obtain CuNi@O-C=O (CN@EDL).

5. Can the loading amount of carbonyl or ester groups on CuNi be controlled precisely? What's the loading amount of the groups in the cocatalysts used in this work? More characterizations should be provided here.

Response: Thank you very much for your comments. We believe that this is a meaningful question. The content of functional groups may indeed affect the performance of CuNi, but we have not strictly controlled the content of functional groups in this study. The loading amount of functional groups may be controlled by adjusting the amount of organic compounds in the precursor or the reaction time. However, this is not the focus of this paper due to the limitation of the length of the manuscript. We were focused on the idea that introduce the introducing electronegative molecules on the surface of CuNi to construct the EDL with a negative electrode outside will optimize the work function, Fermi level, Gibbs free energy of hydrogen adsorption, d-band center, ect. We think your opinion is a good research direction, and we will continue to study this topic in depth.

6. Work functions of other cocatalysts, such as CuNi@O-C=O should be added in Figure S2.

Response: Thank you very much for your comments. We have calculated the work functions of CdS, CuNi, CuNi@C=O and CuNi@O-C=O respectively to analyze the effect of functional groups on the work function and Fermi level. The work function of CuNi@O-C=O is shown in Fig. 1a in the manuscript, while the work functions of CdS, CuNi and CuNi@C=O are shown in **Supplementary Figure 2**.

Supplementary Figure 2 Work functions of (a) CdS, (b) CuNi and (c) CuNi@C=O, respectively.

Fig. 1 | DFT calculation results of electronic characteristics and EDL mechanism. a, Work function of CuNi@O-C=O.

7. The intensities differences in light absorption and steady-state PL spectra are not convincing as the samples amount may be different.

Response: Thank you very much for your comments. We think your suggestion is very meaningful. Indeed, the light absorption intensity and PL intensity is related to the amount of samples used in the test, which is also a problem often ignored by many studies. Therefore, during the UV-Vis absorption and PL tests, we have controlled that the quality of each sample is the same (for example, 200 mg).

The above is our response to the comments, and the specific modifications are shown in the revised manuscript and SI file. Please don't hesitate to contact us if you have any comment about our revised manuscript. We are looking forward to receiving your valuable comments to make the manuscript more perfect. And thank you once more for your valuable comments!

Sincerely,

Chengxin Zhou

Response letter to reviewer 4

Dear reviewer,

Thank you very much for your meaningful suggestions, we have learned a lot from your points. According to your comments, we have made detailed modifications and made responses to each question. The specific contents are as follows:

[The authors describe a novel strategy to boost the photocatalytic performance of the catalysts by coating the CuNi cocatalyst with electronegative molecules. The coating introduces an EDL, which is responsible for the enhanced photocatalytic performance of the catalysts. However, the DFT calculations in the manuscript are quite simplified and can not sufficiently support the idea. And the experimental evidence is vague and lacks a determining proof. Furthermore, the figures in the SI are not quite high in quality. Some critical concerns should be addressed before its publication in this or other journals. Based on the facts mentioned above, I do not suggest its publication in Nature Communications.]

Response: Thank you very much for taking the valuable time to guide our research and put forward meaningful suggestions. We have learned a lot from your comments.

1. The models used in this research are quite simplified and can not support the core idea of the authors. The lengths of the model in the x and y directions are quite small and cannot eliminate the interactions from the adjacent supercell, especially those between the functional groups. Due to the quantum description of the functional groups, could the authors discuss the difference between the ester group and the carbon dioxide

molecules? From my point of view, the functional group that is put on the surface of the CuNi is quite the same as the CO₂ molecules. So, would the CO₂ or CO molecules work similarly and can generate the EDL? Moreover, the models in Figure S1 indicate the presence of two functional groups, why? Would they influence each other? A caption indicating the species of the atoms should be added in Figure S1. The distance between the carbon atoms that bond to the metal atoms should be provided to determine whether a chemical bond forms. Furthermore, different configurations of the functional groups adsorbed on the surfaces should be compared to find the optimum one. Overall, the DFT models should be carefully checked and modified to satisfy these requirements.

Response: Thank you very much for your comments. These opinions are of great significance for in-depth analysis of the problem. Next, we will respond to all your questions separately.

(1) For the problem that the models are quite simplified, we tried to calculate a more bigger supercell system at the beginning of this study, but it took a quite long time, so we simplified the system reasonably (for example, 40 Cu atoms and 40 Ni atoms are used when we calculate the charge density difference and Bader charge) and only part of the model is shown in the manuscript. Under this condition, the calculation of the system took more than half a year.

(2) For the problem that discuss the difference between the ester group and the carbon dioxide molecules, we have set the bond length and bond angle according to the structure of ester group when modeling, as shown in **Fig. 2b**, which is formed by a C-

O and a C=O bond. However, CO₂ is a linear molecule formed by two C=O bonds with a bond angle of 180°. Therefore, the structure of ester group is different from CO₂.

Fig. 2 | b, Angle and bond length in ester group.

Molecular model of CO₂.

(3) For the problem that there are two functional groups in the models in Figure S1, only partly of the model was shown in the manuscript and put two functional groups in the model is to show that functional groups can be distributed on both Cu and Ni.

(4) For the problem that a caption indicating the species of the atoms should be added in Figure S1: we are very sorry that we forgot to add the caption indicating the species of the atoms in Figure S1. It has been added as shown in the revised **Supplementary Figure 1** and thank you very much for your reminder.

(5) For the problem that the distance between the carbon atoms that bond to the metal atoms should be provided to determine whether a chemical bond forms: the distance between the carbon atoms that bond to the metal atoms are measured to be 2.474 Å and 2.737 Å for Cu-C and Ni-C respectively, which are larger than that of the Cu-C and Ni-C chemical bond lengths (1.8~2.0 Å). This proves that the functional group is attached to the surface of CuNi alloy in the form of free radicals, rather than forming Cu_xC compound. We have supplemented this result in the revised manuscript.

*DFT calculations were performed firstly to demonstrate the conjecture that electronegative moleculars (such as $-C=O$ and $-O-C=O$) can regulate the work function, H adsorption/desorption ability, d-band center, etc. The slab models are described in **Supplementary Fig. 1**. The distance between the carbon atoms that bond to the metal atoms are measured to be 2.474 Å and 2.737 Å for Cu-C and Ni-C respectively, which are larger than that of the Cu-C and Ni-C chemical bond lengths (1.8~2.0 Å). This proves that the functional group is attached to the surface of CuNi alloy in the form of free radicals, rather than forming Cu_xC compound.*

(6) For the problem that different configurations of the functional groups adsorbed on the surfaces should be compared to find the optimum one: At first, we built different models, and then carried out structural optimization to screen out the most stable and reasonable configuration. Then combined with the experimental data, the subsequent theoretical calculation is carried out based on this model. Therefore, the model shown

in this paper has been optimized and is considered to be the relatively stable and reasonable structure.

The above is our response to these questions, and we have checked the model to satisfy the requirements according to your suggestions. Thank you very much for your comments, if there is anything inappropriate, please do not hesitate to contact us.

2. The authors proposed a possible working principle of the EDL. However, limited evidence is provided to support this mechanism. The authors are suggested to offer both experimental and theoretical evidence to demonstrate the possible principles.

Response: Thank you very much for your comments. We think this is a very meaningful question. We have also considered using experimental methods to visually test EDL, but found it difficult to directly measure the electric-double-layer or built-in electric field. In order to analyze the formation of the EDL, charge density difference and Bader charge were calculated. By analyzing the calculated results of the charge density difference, we found that the electrons migrated from CuNi to the ester group. Moreover, combining the calculation results of Bader charge, we found that the Cu and Ni atoms did lose electrons, on the contrary, the O atom gained electrons, which is consistent with the calculation results of charge density difference. Therefore, it can be concluded that the electrons tend to gather on the electronegative ester groups to form an additional polarization field pointing to the outside, and thus form the EDL with negative electrode outside.

3. The authors are suggested to cite the classic work of Nørskov et al. to honor their contribution to the d-band theory.

Response: Thank you very much for your comments. We have carefully studied the classic works of Professor Jens Kehlet Nørskov in the field of d-band theory and theoretical calculation, and cited them in the manuscript.

[51] J. K. Nørskov, F. Studt, F. Abild-Pedersen, T. Bligaard, *Fundamental Concepts in Heterogeneous Catalysis*, John Wiley & Sons, New York 2014.

[52] J. K. Nørskov, F. Abild-Pedersen, F. Studt, T. Bligaard, *Proc. Natl. Acad. Sci. U. S. A.* 108 (2011) 937.

[53] B. Hammer, Y. Morikawa, J. K. Nørskov, *Phys. Rev. Lett.* 76 (1996) 2141.

[54] J. K. Nørskov, T. Bligaard, J. Rossmeisl, C. H. Christensen, *Nat. Chem.* (1) 2009, 37.

[55] J. K. Nørskov, T. Bligaard, J. Rossmeisl, C. H. Christensen, *Towards the computational design of solid catalysts*, *Nat. Chem.* 1 (2009), 37–46.

[56] B. Hammer, J. K. Nørskov, *Theoretical surface science and catalysis—calculations and concepts*. *Adv. Catal.* 45 (2000), 71–129.

[57] A. Vojvodic, J. K. Nørskov, *New design paradigm for heterogeneous catalysts*. *Nat. Sci. Rev.* 2 (2015) 140–143.

4. No citations can be found except in the Introduction section. The authors are advised to check the manuscript carefully and provide the necessary some.

Response: Thank you very much for your comments. It is our negligence to leave out the citations in the **Results** section. Thank you very much for pointing out our mistakes. We have adjusted the citations in this section, as shown in the revised manuscript.

For example:

Consequently, the CuNi@O-C=O possesses larger work function and lower Fermi level should capture electrons more easily.^[13]

*as depicted in **Supplementary Fig. 5**, meanwhile the position of the formed anti-bond orbital is paramount for the stability of the material.^[7, 51-54]*

The ΔG_H of Ni-O₂ is closest to zero (-0.039 eV, even better than Pt, $\Delta G_{H(Pt)}=-0.22$ eV)^[9].

On the basis of the DFT results, the cocatalytic performance of CuNi will realize substantial improvement by introducing the ester group to form an EDL in terms of increasing the work function and H adsorption/desorption ability, reducing the Fermi level and d-band center.^[9, 13, 19, 36, 45]

Since CdS mainly exposes the (001) facet, the crystal will grow along [1-100] and the symmetry directions.^[1, 10, 27]

5. Via a simple hydrothermal method, the CN@EDL can be obtained. So where does the O-C=O come from? A detailed investigation of the formation process of this unique structure should be carried out. What is the exact form of the O-C=O? As a molecule or as a free radical? A schematic illustration would be fine.

Response: Thank you very much for your comments. We think this is a very meaningful

question. In this preparation process, the CuNi complex compound was first synthesized by hydrothermal and solvothermal method; secondly, the functional group was obtained by high-temperature calcination due to the decomposition of organic components in the precursor. Therefore, the ester group comes from organic substances such as polyethylene glycol in the precursor. The organic functional group on the surface of CuNi was identified as ester group by XPS, FT-IR and Raman spectra. The ester groups are considered to be exist in the form of free radicals adsorbed on the surface of CuNi as the distance between the carbon atoms that bond to the metal atoms are measured to be 2.474 Å and 2.737 Å for Cu-C and Ni-C respectively, which are larger than that of the Cu-C and Ni-C chemical bond lengths (1.8~2.0 Å). This proves that the functional group is attached to the surface of CuNi alloy in the form of free radicals, rather than forming Cu_xC compound.

The exact structure of ester group is shown in **Fig. 2b** (or response letter **Page 39**).

Fig. 4 | Functional group and interfacial interaction analysis. High-resolution XPS spectra of **(a)** C of CN@EDL. **(b)** FT-IR and **(c)** Raman spectra of CN@EDL.

The above is our response to this question. If you feel there are still some deficiencies, please do not hesitate to contact us.

6. How the EDL promotes photocatalytic activity? A detailed discussion is needed to clarify the role it plays. What is the STH efficiency for this unique structure? A comparison with the recently reported ones in terms of the STH efficacy and activity should be provided to prove its advantage.

Response: Thank you very much for your comments. Firstly, the EDL can improve the carrier dynamics due to the formation of polarization field, as the polarization field would weaken the escape of electrons, thereby increasing the potential barrier of electrons transition from Fermi level to Vacuum level, thus lead to the increasement of work function, which will eventually enhance the ability of the cocatalyst to capture electrons. Moreover, the decreased hydrogen evolution overpotential and effective carrier transfer channel arose by EDL are essential for interfacial carrier kinetics and surface reaction dynamics. Secondly, the experimental results show that CN@EDL is working as the electron acceptor and is the electron-rich site in the composite, therefore, the thermodynamics is optimized by regulating the chemical coordination of surface atoms.

Since this study is not focus on photocatalytic overall water splitting, therefore, the STH calculation has not been carried out previously as the STH calculation is only applicable to the overall water splitting system. According to your suggestion, we carried out the test of overall water splitting hydrogen production under standard AM1.5G sunlight and supplemented the test and calculation method of STH in the manuscript. Results show that the STH efficiency for this structure is about 3.83% (20 mg). However, the calculation of STH is closely related to the amount of catalyst used

in the test. The amount of catalyst used in different documents is different, such as 100 mg, 300 mg, etc., and the larger the amount of catalyst, the greater the SHT value.

To achieve the apparent quantum efficiency (AQE), band-pass filter (PLS-BP20420, FWHM=21 nm) was employed under a 300W Xe lamp (Perfect Light PLS-SXE300). For the measurement of Solar-to-Hydrogen efficiency (STH), the standard AM1.5G filter was employed under a 300W Xe lamp (Perfect Light PLS-SXE300).

The STH efficiency can be calculated by Equation 6:

$$STH = \frac{R(H_2) \times \Delta Gr}{P_{sun} \times S} \times 100\% \quad (6)$$

Where $R(H_2)$ is the photocatalytic hydrogen evolution rate (20 mg, mmol s^{-1}), ΔGr is the molar Gibbs free energy of water splitting reaction (237 kJ mol^{-1}), P_{sun} is the optical power density of the AM1.5G standard solar spectrum (100 mW cm^{-2}), S is the irradiation area (19.6 cm^2), the STH efficiency is calculated to be 3.83%.

Fig. 1 | DFT calculation results of electronic characteristics and EDL mechanism. a, Work function and **b,** the partial density of states (PDOS) of CuNi@O-C=O. **c,** Charge density difference diagram of CuNi@O-C=O, where cyan indicates that the charge density decreases and yellow indicates that it

increases. **d**, The calculated Bader charge of CuNi@O-C=O. **e**, Working mechanism of the EDL.

7. The authors claim that the catalysts “remained stable after more than 300 days”. Does the photocatalytic activity remain stable? If so, prove it.

Response: Thank you very much for your comments. The photocatalysts have been placed under environmental conditions for one year, and **we have tested the** XRD spectra of the fresh sample and stored for 300 days (**Supplementary Figure 34**), as well the XPS tests (**Supplementary Figure 35**), the results show that their structure and composition have not changed, neither photocorrosion nor oxidation, so we claim that the catalysts “remained stable after more than 300 days”. According to your suggestion, we tested the hydrogen production performance of the sample placed for more than one year, and the results showed that there was no obvious change, indicating that its photocatalytic performance remained stable.

8. What do the symbols in the equations in the Methods mean? The authors are strongly suggested to provide the information.

Response: Thank you very much for your comments. Because there are too many symbols involved, we have listed the meaning of some symbols in the **Supporting Information (Supplementary Table 8 and 9)**. The meaning of the symbols in each equation is as follows:

Equation 1: We have added the meaning of symbols in Kubelka-Munk equation

(Equation 1).

The band gaps of CdS and CuNi@O-C=O/CdS are obtained according to the Kubelka-Munk equation (**Equation 1**):

$$\alpha hv = A(hv - E_g)^{1/2} \quad (1)$$

where α is the absorption coefficient, $h\nu$ represents the photon energy (h is Planck constant and ν is the frequency of light), E_g is the energy band gap value.

Equation 2 and Equation 3:

The meanings and values of parameters in **Equation 2** and **Equation 3** are shown in **Supplementary Table 8 (SI file Page 48)**.

Equation 4:

The relationship between E_{CB} , E_{VB} and E_g is shown in **Equation 4**:

$$E_{VB} = E_{CB} + E_g \quad (4)$$

where E_{VB} is the valence band potential, E_{CB} is the conduction band potential, E_g is the energy band gap value.

Equation 5:

The AQE can be calculated by **Equation 5**, and the meaning and value of each parameter is shown in **Supplementary Table 9 (SI file Page 49)**.

Equation 6:

The STH efficiency can be calculated by **Equation 6**:

$$STH = \frac{R(H_2) \times \Delta G_r}{P_{sun} \times S} \times 100\% \quad (6)$$

Where $R(H_2)$ is the photocatalytic hydrogen evolution rate (20 mg, mmol s⁻¹), ΔG_r is the molar Gibbs free energy of water splitting reaction (237 kJ mol⁻¹), P_{sun} is the optical

power density of the AM1.5G standard solar spectrum (100 mW cm^{-2}), S is the irradiation area (19.6 cm^2), the STH efficiency is calculated to be 3.83%.

9. The authors are advised to pay attention to the aesthetics of the Figures. The font of the text and the length of the scale bar in Figure 3 is complete chaos. The discrepancy in font size, font, and position of the caption in the Figures in the SI are quite common. The author should pay more attention to these problems. Furthermore, the authors are strongly suggested polishing their manuscript to avoid some possible typos or grammatical mistakes.

Response: Thank you very much for your comments. Thank you very much for pointing out our mistakes. According to your comments, we have carefully revised the font of the text and the length of the scale bar in **Figure 3**. However, due to the small size of some images, such as **Figures 3e, f**, we have appropriately reduced the font size. The font size, font, and position of the caption in the Figures in the SI file were also carefully checked and revised. Some pictures in the SI file have been rearranged and enlarged to make their display clearer, such as **Supplementary Figure 2-5, 8, 9, 10-15, 17, 18, 24, 28, 30-32, 35-40**. We have supplemented the contents indicated by the arrow in **Supplementary Figure 10** and added more details of the fig captions, as shown in the revised manuscript and SI file (marked in red). What's more, we have carefully checked the grammatical of the manuscript according to your suggestion.

The above is our response to the comments, and the specific modifications are

shown in the revised manuscript and SI file. Please don't hesitate to contact us if you have any comment about our revised manuscript. We are looking forward to receiving your valuable comments to make the manuscript more perfect. And thank you once more for your valuable comments!

Sincerely,

Chengxin Zhou

REVIEWERS' COMMENTS

Reviewer #2 (Remarks to the Author):

The revision has addressed my comments well and I only have a few additional comments:

1. The author should give the full spelling of the words HER, PTC, SEM, XRD, SAED, HAADF-STEM, etc.
2. The two paragraphs describing XPS analysis in Page 10 can be merged into one paragraph.
3. In page 10, line 191, is O=C supposed to be O-C=O?
4. In page 13, the authors claimed that “xxx, indicating that the photocorrosion of CdS was effectively inhibited after loading CN@EDL.”, is there any explanation or literature support?

Reviewer #3 (Remarks to the Author):

The manuscript can be considered the publication in Nature Communications after the following revision.

1. The author claimed that “Based on theoretical simulation, we designed CuNi@EDL (CN@EDL) and applied it as the cocatalyst of semiconductor photocatalysts, finally achieved a hydrogen evolution rate of 249.6 mmol h⁻¹ g⁻¹ and remained stable after more than 300 days.” This description has a property of temporary ambiguity and should be corrected. As the catalyst is only stored under environmental conditions for 300 days. Moreover, 15h for the stability test in photocatalytic reaction is too short. Long-term H₂ production test should be given.
2. The mechanistic study on the high H₂ evolution rate is still ambiguous. For example, why the EDL can enhance the light absorption?

Reviewer #4 (Remarks to the Author):

Comments to NCOMMS-22-47760A

The quality of the manuscript is significantly improved. However, some simple issues should be addressed before publication in Nature Communications, as listed in the following.

Q1: The models used to find out the most stable configurations and corresponding energies should be provided in the SI.

Q2: The authors argue that the ester group forms as the result of high-temperature calcination at 500 °C. However, more reports indicate that the treatment leads to the formation of carbon. How would the authors explain this. Is there any possibility exists that the C=O group comes from the produced carbon materials? Maybe DSC results would be helpful.

Q3: The authors are strongly suggested to provide the photocatalytic performance of the sample after 300 days in the SI or the manuscript.

Response letter to reviewer 2

Dear reviewer,

Thank you very much for your meaningful suggestions, we have learned a lot from your points. According to your comments, we have made detailed modifications to the manuscript and responded to each question. The specific contents are as follows:

[The revision has addressed my comments well and I only have a few additional comments:]

Response: Thank you very much for your positive affirmation of our article. We have learned a lot from your comments.

1. The author should give the full spelling of the words HER, PTC, SEM, XRD, SAED, HAADF-STEM, etc.

Response: Thank you very much for your comments! The full spelling annotations for most abbreviations are in the “Methods” section. We are very sorry for missing the full spelling of some abbreviations. We have supplemented them, as shown in the revised manuscript (marked in red).

2. The two paragraphs describing XPS analysis in Page 10 can be merged into one paragraph.

Response: Thank you very much for your comments. We have merged the two paragraphs describing XPS analysis into one to make them more complete.

3. In page 10, line 191, is O=C supposed to be O-C=O?

Response: Thank you very much for your comments. It was our negligence that made a writing error. Thank you very much for pointing out our mistakes, which has been

corrected in the revised manuscript.

4. In page 13, the authors claimed that “xxx, indicating that the photocorrosion of CdS was effectively inhibited after loading CN@EDL.”, is there any explanation or literature support?

Response: Thank you very much for your comments. In previous studies, it has been found that CdS is prone to photocorrosion, as the S^{2-} is easily oxidized by photogenerated holes. For example, we tested the XRD and XPS of the single CdS after photocatalytic reaction and found the formation of high valent sulfates (SO_4^{2-}), as shown in **Supplementary Figure 27** and **28**. However, by analyzing the XRD and XPS of the CN@EDL/CdS systems that underwent long-term photocatalytic reactions, it was found that there was no obvious change compared with the fresh sample (**Supplementary Figure 29-32**), so we concluded that the photocorrosion of CdS has been inhibited after loading CN@EDL.

The above is our response to the comments, and the specific modifications are shown in the revised manuscript and SI file. Please don't hesitate to contact us if you have any comment about our revised manuscript. We are looking forward to receiving your valuable comments to make the manuscript more perfect. And thank you once more for your valuable comments!

Sincerely,

Chengxin Zhou

Response letter to reviewer 3

Dear reviewer,

Thank you very much for your meaningful suggestions, we have learned a lot from your points. According to your comments, we have made detailed modifications to the manuscript and responded to each question. The specific contents are as follows:

[The manuscript can be considered the publication in Nature Communications after the following revision.]

Response: Thank you very much for your affirmation of our work and your valuable comments!

1. The author claimed that “Based on theoretical simulation, we designed CuNi@EDL (CN@EDL) and applied it as the cocatalyst of semiconductor photocatalysts, finally achieved a hydrogen evolution rate of $249.6 \text{ mmol h}^{-1} \text{ g}^{-1}$ and remained stable after more than 300 days.” This description has a property of temporary ambiguity and should be corrected. As the catalyst is only stored under environmental conditions for 300 days. Moreover, 15h for the stability test in photocatalytic reaction is too short. Long-term H₂ production test should be given.

Response: Thank you very much for your comments. This statement “remained stable after more than 300 days” is not very accurate and prone to ambiguity, so we have corrected it as “remained stable after storing under environmental conditions for more than 300 days” according to your suggestion. About the stability testing, considering the safety of the experiment and the lifespan of the Xenon lamp, we conducted multiple days of testing on the same sample. For example, after completing 15 hours of testing,

we stopped testing and continued to use the same sample from scratch the next day. In fact, the testing time for each sample exceeded 15 hours and remained basically stable. However, in order to express it more rigorously, we have expressed that its performance is stable for at least 15 hours.

2. The mechanistic study on the high H₂ evolution rate is still ambiguous. For example, why the EDL can enhance the light absorption?

Response: Thank you very much for your comments. We are very sorry that our expression here is not rigorous enough. As shown in **Fig. 5c**, the light absorption intensity of CN@EDL is noticeable higher than that of CdS. The absorption intensity of CdS in the range of 530 to 800 nm is virtually zero, after loading CN@EDL, the absorption intensity of CdS system is enhanced in the range of 200 to 800 nm. The improvement in light absorption intensity is mainly due to the introduction of CN@EDL with better light absorption performance. EDL mainly affects the kinetic of electrons and local thermodynamic properties.

The above is our response to the comments, and the specific modifications are shown in the revised manuscript and SI file. Please don't hesitate to contact us if you have any comment about our revised manuscript. We are looking forward to receiving your valuable comments to make the manuscript more perfect. And thank you once more for your valuable comments!

Sincerely,

Chengxin Zhou

Response letter to reviewer 4

Dear reviewer,

Thank you very much for your meaningful suggestions, we have learned a lot from your points. According to your comments, we have made detailed modifications to the manuscript and responded to each question. The specific contents are as follows:

[The quality of the manuscript is significantly improved. However, some simple issues should be addressed before publication in Nature Communications, as listed in the following.]

Response: Thank you very much for your encouragement and affirmation of our work, and for your valuable comments!

1. The models used to find out the most stable configurations and corresponding energies should be provided in the SI.

Response: Thank you very much for your comments! In the initial modeling process, we tried different models (for example, there is a layer of amorphous carbon or graphene on the surface of CuNi alloy, and then functional groups attach to the surface of graphene, etc), but during structural optimization, we found that some models could not achieve structural convergence, or after completing structural optimization, we compared the atomic positions in CONTACAR and POSCAR and found that there were significant changes, which completely did not meet the expectations. Therefore, we believed that these configurations may be unreasonable, so we abandoned these models. Further combined with characterization results such as XRD, no characteristic peaks of carbon or graphene were found. Moreover, C-C is electrically neutral, and does not

affect the electric double layer properties in this system. At the beginning of this study, we proposed to introduce electronegative molecules to build an electric double layer (EDL) to generate a polarization field to improve carrier dynamics, and optimize the thermodynamics by regulating the chemical coordination of surface atoms. Therefore, introducing electronegative molecules on the surface of CuNi to construct the EDL with a negative electrode outside was proposed.

2. The authors argue that the ester group forms as the result of high-temperature calcination at 500 °C. However, more reports indicate that the treatment leads to the formation of carbon. How would the authors explain this. Is there any possibility exists that the C=O group comes from the produced carbon materials? Maybe DSC results would be helpful.

Response: Thank you very much for your comments! We agree with your opinion that there will be carbon in this system, but the specific form of carbon is difficult to determine, such as amorphous carbon or graphene (according to experience, amorphous carbon is likely to be formed at this temperature), while amorphous carbon is relatively complex and diverse, which is difficult to model scientifically. Based on your comments, we have tested TG and DSC and found that the quality of the material still changes when calcined to 500-900°C, indicating that the residual product at this temperature should be the decomposition product of organic matter. During the calcination process, a portion of carbon may be formed, and as the calcination temperature increases, it may even gradually carbonize. Therefore, we controlled the calcination temperature not to be too high to reduce the carbon content formed. Moreover, the existence forms of

carbon and carbon oxygen groups in the system may be very complex and diverse, with some groups interacting with carbon and others directly acting on metal atoms. Researches have shown that carbon layers are very important for enhancing conductivity, but in order to make the focus of this work more prominent, we have focused on the impact of functional groups mediated polarization field on carrier dynamics and thermodynamics. Furthering considering that C-C is electrically neutral, and does not affect the electric double layer properties in this system.

The above is our explanation of this question, and we fully agree with your viewpoint. It is also a problem that we have analyzed and discussed for a long time during the research process. Finally, considering all aspects, we have adopted this analysis method without affecting the central idea.

3. The authors are strongly suggested to provide the photocatalytic performance of the sample after 300 days in the SI or the manuscript.

Response: Thank you very much for your comments. We are very sorry that we did not include the specific hydrogen production rate of the sample after 300 days in the article. We have added it, as shown in **Supplementary Figure 34**.

Supplementary Figure 34 Hydrogen production rates of (a) fresh and (b) stored for 300 days of CN@EDL(5%)/CdS.

It can be seen that compared with fresh sample, the hydrogen production activity of CN@EDL(5%)/CdS stored for more than 300 days did not significantly decrease, indicating that the photocatalyst is stable under environmental conditions. The slight difference in hydrogen production rates may come from the inherent errors in different batches of testing, as well as the gradual decrease in the light power density of xenon lamps with increasing usage time.

The above is our response to the comments, and the specific modifications are shown in the revised manuscript and SI file. Please don't hesitate to contact us if you have any comment about our revised manuscript. We are looking forward to receiving your valuable comments to make the manuscript more perfect. And thank you once more for your valuable comments!

Sincerely,

Chengxin Zhou